# SAPEVO-H² a Multi-Criteria Systematic Based on a Hierarchical Structure: Decision-Making Analysis for Assessing Anti-RPAS Strategies in Sensing Environments

Miguel Ângelo Lellis Moreira [1,2,*], Fernando Cesar Almeida Silva [1], Igor Pinheiro de Araújo Costa [1,2], Carlos Francisco Simões Gomes [1] and Marcos dos Santos [2,3]

1    Production Engineering Department, Federal Fluminense University, Rio de Janeiro 24210-240, Brazil
2    Operational Research Department, Naval Systems Analysis Centre, Rio de Janeiro 20091-000, Brazil
3    Systems and Computing Department, Military Institute of Engineering, Rio de Janeiro 22290-270, Brazil
*    Correspondence: miguellellis@hotmail.com

**Abstract:** Regarding high-level and complex decision-making scenarios, the study presents an extensive approach to the Simple Aggregation of Preferences Expressed by Ordinal Vectors-Multi Decision Making method (SAPEVO-M). In this context, the modeling proposal, named SAPEVO-Hybrid and Hierarchical (SAPEVO-H²), the objective of this study, based on the concepts of multi-criteria analysis, provides the evaluation of alternatives under the light of multiple criteria and perceptions, enabling the integration of the objectives of a problem, which are transcribed into attributes and structured in a hierarchical model, analyzing qualitative and quantitative data through ordinal and cardinal entries, respectively. As a case study, a decision analysis concerning the defense strategies against anti-Remotely Piloted Aircraft Systems (RPAS) strategies for the Brazilian Navy is carried out. Using the technique of the causal maps approach based on Strategic Options Development and Analysis (SODA) methodology, the problematic situation is structured for numerical implementation, demonstrating the performance of objectives and elements of a hierarchical structure. As a result, rankings concerning objectives and anti-RPAS technologies, based on the treatment of subjective information, are presented. In the end, the main contribution of the study and its limitations are discussed, along with the conclusions and some proposals for future studies.

**Keywords:** anti-RPAS strategies; multi-criteria; hierarchical model; SAPEVO-M method

## 1. Introduction

Based on the Brazilian Federal Constitution [1], it is necessary for the Brazilian Navy (BN) to enable, prepare and employ naval power to contribute to national defense, providing the guarantee of constitutional powers, ensuring law and order in the country, taking into account the duties provided by law, focusing on those related to the Maritime Authority and seeking to contribute to and safeguard national interests [2].

As discussed in the Brazilian National Defense Policy [3], even in times of peace, the armed forces must develop defense strategies to guarantee national sovereignty and security, providing the development and implementation of means, strategic studies that address national security needs and support policy assessment scenarios in the international environment [4]. Strictly according to the BN and based on the National Defense Strategies, they must enable the application of maritime forces in favor of national interests, this being aligned with the strategic and economic policies of Brazil in the international scenario [5].

As discussed in [6], decision making in political and military environments involves different levels and areas, interconnecting strategic, tactical and operational analysis in favor of a direction aligned with the objectives in a given problematic situation [7]. In addition, it is necessary to consider that decision making in the political and military

spheres is complex, where the given form of a solution can generate influences not only in the military sphere but also in other areas of society [8].

In the scenario of high-level decision analysis and the impact in complex environments, integrating multiple stakeholders to determine and analyze aspects relevant to the problem is common [9], enabling, from multiple perspectives, a consensus in decision making [10]. According to [11] with the involvement of multiple scenarios and circumstances, the increase in complexity in a given analysis becomes noticeable, with different points of view as to the importance or influence of a decision variable [12], although it is necessary to consider it in favor of a substantial evaluation and greater assertiveness in the final decision [13].

Considering the scenario of technological advancement that has taken place in recent decades, the improvement of military technologies has provided the use of revolutionary capabilities by military forces [14]. In this context, the Remotely Piloted Aircraft System (RPAS), mainly due to its versatility, has been considered a promising and desirable alternative to traditional flights [15]. The capabilities provided by RPAS range from deploying weapons in distant wars to tracking/monitoring surveillance missions, among many others [16].

However, if, on the one hand, military forces can exploit the operational capabilities allowed for by the use of RPAS, on the other hand, these forces must also be concerned with how to prevent the hostile use of these tools by any opposing forces [17]. In this scenario, strictly for the BN, the need to design anti-RPAS strategies is highlighted, which aligns with the political system of strategic defense worked on by the BN. In a complementary way, the following question is how to provide a feasible conception of anti-RPAS strategies based on the integration of the Strategic, Tactical and Operational objectives established by BN regarding the national remote technology scenario.

The Operational Research (OR), in the context of decision making, enables, through its approaches and methodologies, the analysis of complex, problematic situations with a technical and scientific basis, making it possible to understand the problem and structure and clarify a favorable solution to a given scenario [18]. In addition, it should be noted that the models belonging to OR are not limited to the implementation of a set of equations [19] but rather operate in the development and implementation of algorithms, comprising their axiomatic structure and the integration of logic and mathematics [20]. This is intended for the processing and treatment of data, information and preferences established by the evaluators of a given problematic situation [21,22].

As presented in [23], in real problematic situations of decision analysis, uncertainty variables are intrinsic to the assessment [24], especially in environments involving multiple decision makers belonging to different strategic levels, where members generally disagree about the parameter values and preference assignments [25]. Nevertheless, even though group decisions are expected in analyzing real situations, few models consider the formation of subgroups to deal with problems at strategic, tactical and operational levels [26]; in other words, most of the models do not consider hierarchical analysis, providing the integration of different levels of expertise in the decision-making process. This is a problematic situation and a motivation of the methodology development, which tries to fill this gap regarding group decision-making analysis in complex scenarios.

According to [27], when considering a set of several variables evaluated on multiple attributes, the Multicriteria Decision Support (MCDA) models, which originated in the OR, enable support in the decision-making process, contemplating techniques that allow the decision maker, this being a person, group or organization, to carry out the structuring of variables and preferences, clarifying their respective degrees of importance in an interactive process with other actors [28].

In the above scenario, this study presents a proposal of an extensive approach to the Simple Aggregation of Preferences Expressed by Ordinal Vectors-Multi Decision Making) (SAPEVO-M) [29]. The given systematic, titled SAPEVO Hybrid and Hierarchical (SAPEVO-H$^2$), proposes an integrated assessment of multiple decision makers, enabling

the construction of assessment subgroups intended for the specific analysis of parts of the problematic situation structured in a hierarchical format, which is not restricted to the evaluation of alternatives under multiple criteria but also considers other attributes pertinent to a given problem in a more appropriately strategic relevant scenario in a given situation. In addition to its original model, added to the ordinal evaluation model, the proposed model also enables the treatment of quantitative attributions based on cardinal inputs.

In order to provide the feasibility of the approach proposal, a case study concerning the analysis of anti-RPAS strategies is explored. For a better understanding of the problematic situation, techniques of causal maps based on the principles of Strategic Options Development and Analysis (SODA) [30] are used, enabling the clarification of objectives that are intrinsic to the analysis of strategies, structuring these variables for their treatment and evaluation through the methodological proposal to be presented and performing the integration of multiple decision makers at different hierarchical levels and knowledge of the variables belonging to the problematic situation.

As the main contribution of this study, we intend to provide the problematic situation structuring regarding some of the main objectives of anti-RPAS strategies for the Brazilian Navy, clarifying the priorities between the elements in the evaluation through outranking analysis by the performance of alternatives, serving as an aid in the decision making.

The article is structured into seven sections. After the introduction, Section 2 presents a review of the literature on the application of multi-criteria approaches in the military scope. Section 3 explores theoretical foundations related to MCDA, presenting some models based on evaluations based on hierarchical structures. Finally, the methodology inherent to the SAPEVO-M method is exposed. Section 4 exposes the axiomatic structure of the SAPEVO-H$^2$ modeling, presenting its particularities and evolution points concerning the previous models. Section 5 concentrates on the case study, analyzing the problematic situation and the numerical implementation of the given model, along with analyzing the results. Section 6 briefly discusses the proposed model and the adherence to the application context. Finally, Section 7 presents the study's final considerations based on its results and proposals for future work.

## 2. Decision Making in the Military Scope

Military problematics are of great importance for the world, since their effects and motivations have repercussions for nations' defense and sovereignty scenarios [31]. Highly complex environments, conflicting variables, imprecision in information, subjectivity and uncertainties are some of the main characteristics of real-world problems [32]. In this scenario, MCDA methodologies make decision making more rational and efficient [33,34].

In recent decades, the MCDA methods have provided the structuring, analysis and decision making in the military spheres [35], showing a relative increase with the number of implementations. Much of this fact is due to the feasibility of analyzing sensitive issues that affect the defense issues of the nation [36]. Whenever there is a need to acquire military equipment, such as military training aircraft, armaments or war tanks, many variables and factors are considered in a given assessment of this case; the application of approaches based on the MCDA proves to be of great value as a form of decision support [37].

As explained in [38], the application of models based on the MCDA in the Armed Forces is essential for providing greater accuracy and transparency regarding decision factors, making it possible to reduce resources and increase the defense capacity and assertiveness in the final decision [39]. Regarding the MCDA methods, Santos, Costa and Gomes [40] present the Analytic Hierarchy Process (AHP) method as the model with the most remarkable presence in military-related problems, having, as an example, the allocation of installations in military bases [41]; the evaluation of naval tactical missile systems [42]; the dimensioning of US wrecked fleets [43]; the allocation of resources for the development of anti-terrorism strategies [44]; the ordering and analysis of weapons systems [45]; the analysis of simulation systems [46,47]; the determination of strategies

to support the Global Maritime Fulcrum [48]; and the selection of advanced military training aircraft [49].

Additionally, other MCDA models have also been implemented in other studies of military scopes, such as the TOPSIS method [50–53]; the BORDA method, which is one of the first models based on the MCDA [54,55]; and the MAUT method [56–58]. In addition to these models, studies focused on outranking models were also explored, such as the PROMETHEE [59–62], and ELECTRE [59,63].

Finally, it also identified the application of hybrid models, that is, models that present, in their axiomatic base, the integration of two or more methodologies, with emphasis on the THOR 1 and 2 methods [64,65]; AHP-PROMETHEE [62]; IFM [66]; PROMETHEE-SAPEVO-M1 [67,68]; and ELECTRE-Mor [35].

## 3. Multi-Criteria Decision Analysis (MCDA)

The decision-making process is integrated into human activity, characterized as analyzing a set of actions to obtain a favorable solution to a given problem [69]. The MCDA provides the structuring and understanding of a problem in complex environments, considering risk and uncertainty [70], helping to clarify favorable solutions to problems of a varied nature, which is made possible through an interactive and transparent process [71].

The methods present in MCDA seek to establish preferences between alternatives [72] under the analysis of criteria that often conflict with each other [73]. As discussed in [74], there are commonly three main types of issues addressed in models belonging to AMD, and they are: Choice, exposing the most favorable alternative in a global context; Ordering, establishing order from the most promising alternative to the least favorable ones with a form of the solution; and Classification, allocating the choice into dominance classes [75].

Over the years, numerous MCDA-based decision-making models have been proposed. In this context, two large groups of methods originated [76]: one designated as aggregation methods through a single synthesis/compensatory criterion (American School) and the other classified as outranking/non-compensatory methods (French/European School) [77].

This first group was constructed based on utility theory. In this scenario, it is possible to obtain two types of preference relations, namely, preference relation ($a_iPa_j$) and indifference ($a_iIa_j$). In this specific group, incomparability relations are not considered, and the transitivity between preferences is assumed. Commonly, these methods do not consider uncertainties, imprecision and ambiguity between data [78].

The main methods belonging to this group include the AHP (Analytic Hierarchy Process) [79], MACBETH (Measuring Attractiveness by a Categorical-Based Evaluation Technique) [80,81], MAUT (Multiple Attribute Utility Theory) [82,83], SMART (Simple Multi-Criteria Attribute Rating Technique) [84], TODIM (Portuguese acronym for Interactive Multi-Criteria Decision Making) [85], TOPSIS (Technique for Order of Preference by Similarity to Ideal Solution) [86] and UTA (Additive Utility Theory) [87].

The second group of MCDA methods is characterized by the outranking relation of the alternatives, characterized by the non-transitivity relations between the preferences [88]. The methods belonging to this group extend a basic set of situations of preference relations based on four forms: the indifference relation ($a_iIa_j$), weak preference ($a_iQa_j$), strict preference ($a_iPa_j$) and, finally, the incomparability relation ($a_iRa_j$) [89].

The two main methods that compose this class are considered as families of methods, these being the ELECTRE method (Elimination and Choice Translating Reality for Enrichment Evaluation) [90] and the PROMETHEE method (Preference Ranking Method for Enrichment Evaluation) [91].

### 3.1. MCDA Group Approaches

Decision-making problems in real scenarios are rarely analyzed based on only one influence variable. Therefore, especially when dealing with high-level decision making, it is common to consider the presence of multiple variables that conflict with each other [92]. Additionally, the presence of multiple stakeholders is also common, given the need to

integrate different perspectives and preferences in a given decision-making scenario [93], with the difference between these being important complexity factors that impact the final decision made [94].

The decision-making scenario becomes more challenging and complex with the need for joint assessments or negotiations by multiple decision makers, each with their perceptions, preferences or aspirations for a given problem. In this context, group evaluation methodologies seek to integrate stakeholders' perceptions of a given problematic situation through their participation and interaction, achieving a favorable result within the consensus of multiple evaluators [95].

The search for consensus among the stakeholders of a given analysis is complex in many cases; there is a need for the assistance of a moderating analyst to help the evaluators construct the final decision [96]. Another relevant factor in group assessment scenarios reflects the points of uncertainty involved in the problem, many of which deal with the decision maker's experience and expertise in the problematic situation [97].

As presented in [98], the aggregation of preferences can be obtained by different approaches in a decision-making process. Regarding the MCDA models, there are two ways to implement a given method for group assessment. In the first case, a single form of input can be used, where a given preference represents the group's consensus on a variable, and in the second case, some models enable the clarification of individual preferences and, subsequently, the representation of these in an aggregated format, as presented in [99–102].

In addition to the MCDA methods already established in the literature, other models and extensive approaches have proposed methodological adaptations for group assessments, seeking to treat and integrate the preferences of involved decision makers and evaluators in a given problematic situation [93]. In general, the MCDA models for group assessments are based on three tasks connected to the assessment process; they are the organization and structuring of the decision process, the representation of individual preferences of decision makers and, finally, the aggregation of established preferences [103].

As discussed in [104], the group assessment models integrated into the MCDA can be established into two categories. In the first case, model data are based on evaluation procedures, focusing on the development of interaction between decision makers in favor of clarifying important information, generating new ideas, minimizing disagreement and, finally, determining a favorable alternative, as explored in the models presented in the respective studies [105–109]. In a second context, there are models based on the optimization and aggregation of the final decision, aiming to generate an ideal final decision based on the integration of optimization models, as shown in the studies presented [107,110–113].

### 3.2. The SAPEVO Method

The Simple Aggregation of Preferences Expressed by Ordinal Vectors method (SAPEVO) [114] operates under an ordinal input approach for evaluating variables in a given scenario. Regarding the multi-criteria methodology, a given method enables the processing of subjective and tacit data to the decision maker, thus making it possible to convert these points of subjectivity into cardinal scores, numerically expressing a relative degree of the importance or performance of the criteria and alternatives of decision making.

Over the years, other approaches and methodologies based on the SAPEVO method were developed, enabling an improvement of the model initially proposed in [114]. Providing an analysis composed of multiple decision makers, the Simple Aggregation of Preferences Expressed by Ordinal Vectors-Multi Decision Makers (SAPEVO-M) methodology, proposed in [29], introduces an evaluation format for multiple decision makers in a decision-making process, in addition to the axiomatic improvement of the previously developed model, thus bringing an increase in its consistency.

The model [29] emphasizes that the main characteristic of the model is related to the ordinal transformation process of preferences, which is used to obtain the degrees of preference relations between the alternatives in each criterion and also to obtain the degrees of importance of the criteria, thus generating their respective weights.

Introducing an integration of outranking and classification methodologies from the European school, the PROMETHEE-SAPEVO-M1 [68] and ELECTRE-MOr [115] methods expose hybrid approaches for complex decision evaluation scenarios, providing the analysis of variables of a quantitative and qualitative nature. This is made possible through axiomatic concepts proposed in the SAPEVO-M method [29], integrating and evaluating quantitative and qualitative (subjective) data in a decision-making process.

Finally, a non-compensatory approach was developed for a given model, named SAPEVO-M-NC [116,117], which provides analysis by multiple decision makers through a non-compensatory ordinal evaluation of the set of variables. As it can be recognized as a family of methods, currently, a given group is composed of five methodologies, as shown in Figure 1. In a complementary way, Table 1 shows the technical characteristics of each model.

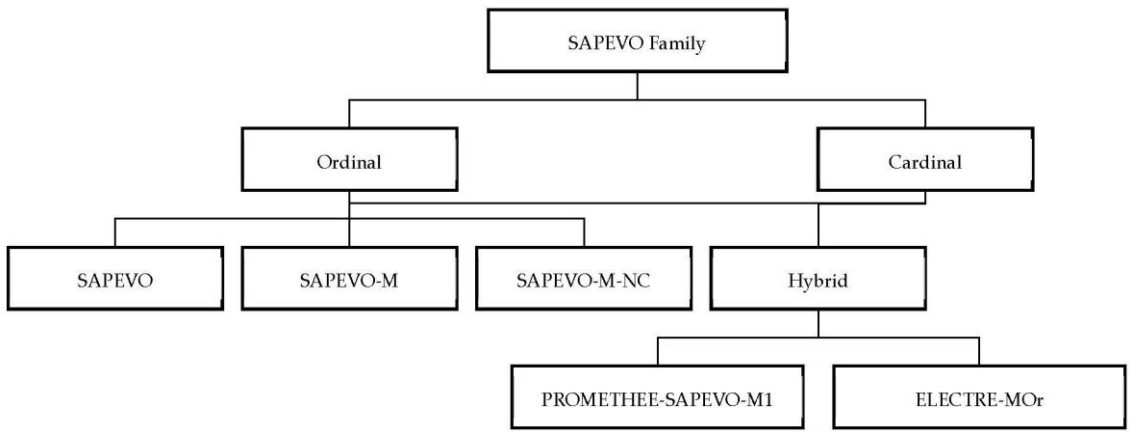

**Figure 1.** Structural distribution of the SAPEVO family methods.

**Table 1.** Technical characteristics of methods related to the SAPEVO family.

| Method | Compensatory | Non-Compensatory | Ordinal | Cardinal | Ranking | Sorting | Group |
|---|---|---|---|---|---|---|---|
| SAPEVO | X | | X | | X | | |
| SAPEVO-M | X | | X | | X | | X |
| PROMETHEE-SAPEVO-M1 | X | | X | X | X | | |
| ELECTRE-MOr | X | | X | X | | X | X |
| SAPEVO-M-NC | | X | X | | X | | X |

A search in the Scopus database identifies 14 papers using at least one method that makes up the SAPEVO family. These studies are in Computer Science, Engineering, Mathematics, Chemistry and Agriculture, among others. It is also noteworthy that, among the models presented, four software have been registered with the National Institute of Industrial Property (INPI).

General Procedure of the SAPEVO Method

The axiomatic process of the model is processed in two parts. Preliminarily, the transformation of the ordinal preference between criteria must be performed and expressed by a vector representing the weights of the criteria. Then, the ordinal transformation of the preference between alternatives within a given set of criteria, expressed by a matrix, is performed. A series of paired comparisons between options, whether criteria or alternatives within a given criterion, denote the individual preference information of each decision maker.

Based on an ordinal scale of seven points (−3, −2, −1, 0, 1, 2, 3), ranging from absolutely worse to absolutely better, pairwise evaluations are carried out, indicating the intensity of preferences between two floats. A given model supports the two parts of the model's analysis; it serves as a basis for evaluating the criteria and the alternatives for

each criterion. The normalization process of variables (alternatives or criteria) allows for the transformation of ordinal scores into cardinal quantities, providing the aggregation of preferences and the ordering of alternatives [29].

## 4. Approach Proposal

The SAPEVO-H$^2$ approach enables the assessment of a given problem by multiple decision makers, allowing each decision maker to evaluate the hierarchy of attributes and criteria partially or totally; a decision maker or a group of decision makers can evaluate all levels of the hierarchy or just part of these levels. It is emphasized that the type of analysis to be carried out reflects the structuring of the problem, which has already been carried out. As shown in Figure 2, the treated modeling consists of four steps divided into sub-steps.

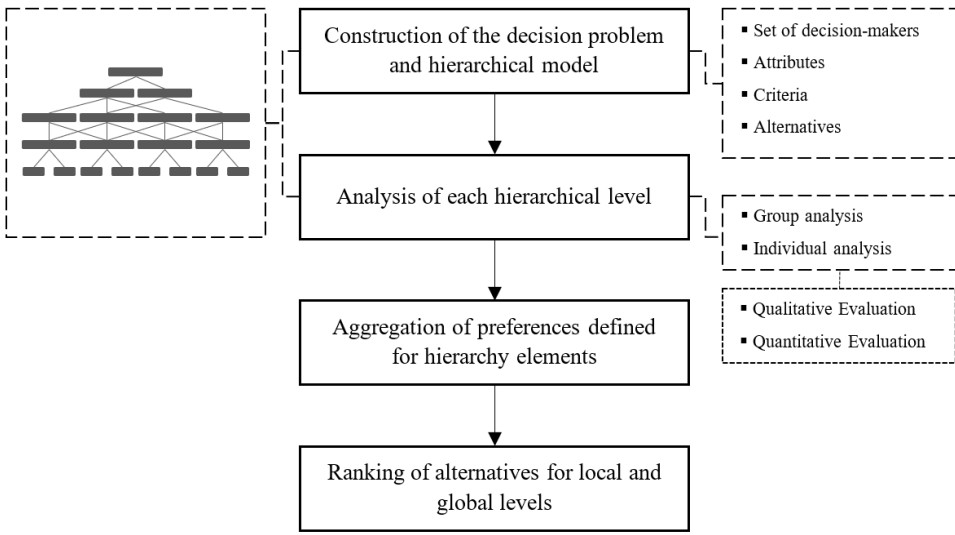

**Figure 2.** Analysis Process SAPEVO-H$^2$.

### 4.1. Construction of the Evaluation Structure

The modeling integrates a set of decision makers $D$, where $d_m \in D$, $m = 1, \ldots n$, enabling the decision maker to evaluate a given problem based on a set of hierarchical levels $N$, where $n_h \in N$, $h = 1, \ldots l$.

Related to the achievement of a given strategic objective, representing the top of the hierarchy under evaluation, for each hierarchical level, a set of elements is determined, which can be:

- Attributes set $T$, where $t_r \in T$, $r = 1, \ldots s$.
- Criteria set $C$, where $c_j \in C$, $j = 1, \ldots k$.
- Sub-criteria set $S$, where $s_u \in S$, $u = 1, \ldots v$.
- Alternatives set $A$, where $a_i \in A$, $i = 1, \ldots b$.

Considering the possible assessment sets, Figure 3 provides an example of a configuration for a given approach, highlighting that, for each set, there is a minimum need for two elements, making it possible to compare the elements belonging to a set at each level of analysis.

### 4.2. Element Analysis

For all sets of evaluations in their respective levels, the evaluation by multiple decision makers or by only one decision maker is made possible. Concerning the analysis of the respective levels of the sets of attributes and criteria, an ordinal scale is used based on seven points, which is shown in Table 2.

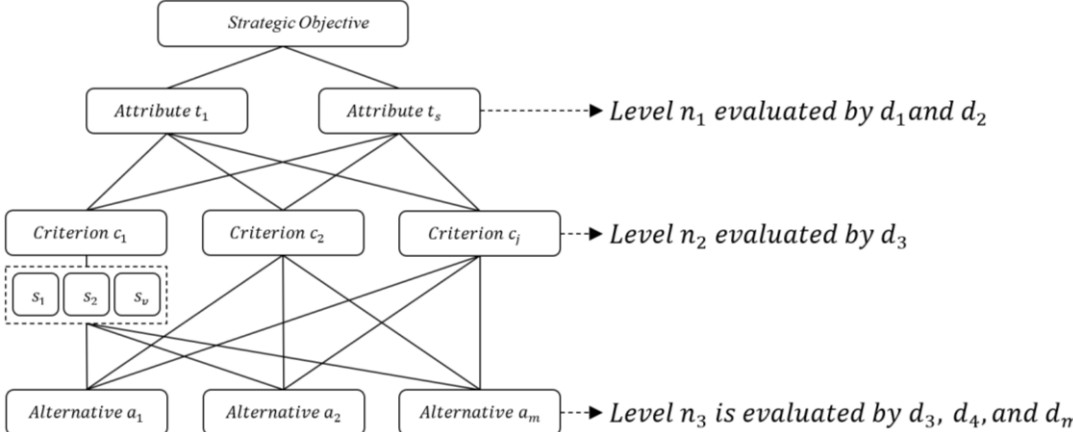

**Figure 3.** Example of a hierarchical structure for the evaluation.

**Table 2.** Ordinal scale.

| Verbal Descriptions | Punctuation |
|---|---|
| Absolutely worse/less important | −3 |
| Much worse/less important | −2 |
| Worst/less important | −1 |
| Equivalent | 0 |
| Best/more important | 1 |
| Much better/more important | 2 |
| Absolutely better/more important | 3 |

### 4.2.1. Ordinal Evaluation with Multiple Decision Makers

For an analysis of a given set of elements with multiple decision makers, the ordinal scale (Table 2) is used for the pairwise assessment between the elements of the set. For each decision maker $d_m$, a matrix with the indications of preferences between the elements of the set is established. Through Equation (1), the sums of the quantities are established, defining a degree of importance for the elements evaluated by a decision maker.

$$v = \frac{\sum a_i - min\ a_i}{max\ a_i - min\ a_i} \tag{1}$$

With the degrees obtained for each element $e_i$, the Sum (2) is performed for each decision maker $m$. Following the procedure, the normalization of the Sums (3) is obtained, indicating the respective importance of the elements for the attribute of the higher level. In this model, keeping the technique indicated in [29], if any criterion or attribute presents zero importance, 1% of the smallest value greater than zero is assigned for this element.

$$e_i = \frac{\sum_{m=1}^{n} e_{im}}{m} \tag{2}$$

$$v_i = \frac{e_i}{\sum e_i} \tag{3}$$

### 4.2.2. Ordinal Evaluation with One Decision Maker

For an analysis by only one decision maker at a given level, an axiomatic variation proposed by Moreira et al. (2021) is used. In this context, a scale based on an upper and lower limit is generated, defined as *maximum sum* (4) and *minimum sum* (5), respectively.

$$maximum\ sum = (n-1)3 \tag{4}$$

$$minimum\ sum = (n-1)(-3) \tag{5}$$

Using the ordinal scale (Table 2), the scores of the elements of a given set under evaluation are obtained, where the normalization of scores is made possible by Equation (6), generating the degrees of importance of the elements. Subsequently, Equation (3) is used to normalize the obtained degrees.

$$v_i = \frac{\sum a_i - minimum\ sum}{maximum\ sum - minimum\ sum} \tag{6}$$

### 4.2.3. Cardinal Evaluation

Considering a hybrid model, the modeling enables not only the evaluation of elements based on qualitative attributions but also quantitative analysis, which is the evaluation of alternatives in quantitative criteria, where the alternatives already have cardinal numerical attributions for each criterion.

For the evaluation, two types of thresholds can be considered, representing each quantitative criterion, a minimum limit $L_{min}$ and a maximum limit $L_{max}$. For each alternative, the degrees of importance $a_{ij}$ are calculated through a set of three preference functions, as presented in Figure 4.

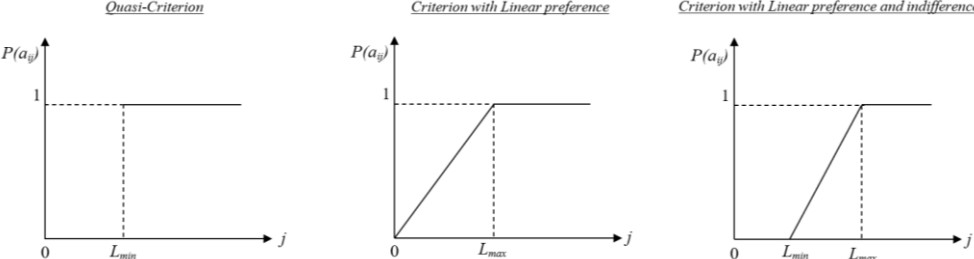

**Figure 4.** Preference functions of cardinal normalization.

The *Quasi-Criterion* function considers only the $L_{min}$ threshold, which sets an indifference point between the variables in the analysis; all variables with a performance up to $L_{min}$ present a strict preference in the criterion, as presented in Equation (7).

$$a_{ij} = \begin{cases} 0\ if\ a_{ij} \leq L_{min} \\ 1\ if\ a_{ij} > L_{min} \end{cases} \tag{7}$$

Regarding the linear function with a preference threshold, a $L_{max}$ threshold is set, where a point of a strict preference is defined, enabling the determination of a linear preference if the value of the variable consists under $L_{max}$. Equation (8) presents the function modeling.

$$a_{ij} = \begin{cases} \frac{a_{ij}}{L_{max}}\ if\ a_{ij} \leq L_{max} \\ 1\ if\ a_{ij} > L_{max} \end{cases} \tag{8}$$

In the third function, it is necessary to set an indifference and preference threshold, where $L_{min}$ is the minimum level for an alternative with some degree in a criterion and $L_{max}$ is the value of strict dominance of a variable in the set. For the performance established between $L_{min}$ and $L_{max}$, a linear evaluation is considered, as presented in Equation (9).

$$a_{ij} = \begin{cases} 0 & if\ a_{ij} \leq L_{min} \\ \frac{a_{ij} - L_{min}}{L_{max} - L_{min}} & if\ L_{min} < a_{ij} < L_{max} \\ 1 & if\ a_{ij} \geq L_{max} \end{cases} \tag{9}$$

All decisions concerning the type of function and the value of thresholds need to be set by the analyst and the decision makers of the process. At the end of the quantitative analysis, for each $a_{ij}$, the Equations (2) and (3) need to be processed.

4.2.4. Consistency Test in Pairwise Evaluation

Considering the attributions performed in the peer-to-peer evaluations, a given axiomatic procedure operates as a way to understand the consistency of the attributions performed in the input matrix.

Taking into account the set of variables in an n-dimensional matrix, a given input is used as a basis for the consistency test model, using its upper diagonal for validation, as shown in Figure 5.

**Figure 5.** Input matrices for testing the consistency, considering variables A, B, C, and D.

As shown in Figure 5, a matrix named as the ideal transitive is constructed, serving as a basis for comparison for a given consistency test. In this context, the values of $a_{ij}$ are obtained by Equation (10).

$$a_{ij} = \begin{cases} -3 & if \ (a_{1j} + a_{i1}) \leq -3 \\ a_{1j} + a_{i1} & if \ -3 < (a_{1j} + a_{i1}) < 3 \\ 3 & if \ (a_{1j} + a_{i1}) \geq 3 \end{cases} \tag{10}$$

Performing the comparison evaluation between the input matrix and the ideal transitive matrix, a new matrix is generated, which is the result of the comparison between the two previous ones, named as the comparison and consistency matrix, which is constituted by binary data {0;1}. The comparison between the two input matrices is carried out, and the difference between the values varies in the range of {−1;1}. $a_{ij} = 0$; if the difference obtained is out of scale, $a_{ij} = 0$. Figure 6 provides an example of a possible matrix.

For each constructed matrix, the sum of the binary scores is obtained, represented by $bp = \sum a_{ij}$. With the value of $bp$, the consistency calculation is performed by the number of assignments performed in a pairwise comparison evaluation, represented by $\frac{n(n-1)}{2}$. With given values, the consistency value $\lambda$ is generated, represented in Equation (11).

$$\lambda = \frac{bp}{\left(\frac{n(n-1)}{2}\right)} \tag{11}$$

Once the consistency value is obtained, Table 3 supports the understanding of the consistency relationship obtained between the comparisons attributed in the pairwise evaluations of the proposed model.

*Comparison and Consistency Binary Matrix*

|   | A | B | C | D | n |
|---|---|---|---|---|---|
| A | - | 0 | 0 | 0 | 0 |
| B |   | - | 0 | 0 | 1 |
| C |   |   | - | 1 | 0 |
| D |   |   |   | - | 0 |
| n |   |   |   |   | - |

**Figure 6.** Consistency Matrix with binary values, considering variables A, B, C, and D.

**Table 3.** Consistency relations.

| Consistency | Percentage |
|---|---|
| High | 0–10% |
| Average | 10–20% |
| Low | 20–30% |
| Inconsistent | 30–40% |
| Very Inconsistent | 40–100% |

*4.3. Outranking Index*

For each evaluated level, an outranking relation is built between the elements belonging to a given level by obtaining the local performance index, represented by Equation (12), where $e_i$ represents the evaluated element and $v_{j(h-1)}$ represents the weighted degree of preference of the respective element at the top level and the $h$ index under analysis. In this context, an outranking matrix $rs$ between the alternatives is generated, providing the construction of the outranking Index (13) of the variables at each level analyzed.

$$\phi^+ e_{ih} = \frac{\sum_{j=1}^n e_{ijh} v_{j(h-1)}}{n} \tag{12}$$

$$\phi e_h = \sum_{r=1}^s \phi^+ e_{rh} - \phi^+ e_{sh} \tag{13}$$

*4.4. Aggregation Process*

Once the degrees of importance of all elements of the hierarchy are assigned, the values are aggregated, indicating the relative importance of the element $e_i$ regarding the element of the upper-level $s_j$ by $e_{ij} = e_i s_j$. For every two levels above a given element $k_l$, its importance is obtained through the additive aggregation model indicated in Equation (14).

$$e_{ijl} = \sum_{l=1}^n e_{ij} k_l \tag{14}$$

With the procedure of additive aggregation, it is possible to obtain the performances of the alternatives, leading to their ranking in a global and local format. As it is a hierarchical model, it is possible to analyze the performance of alternatives in each criterion or attribute, clarifying the global result based on decision makers' attributions at different levels of the hierarchy. It is noteworthy that, in addition to obtaining the performance of the alternatives, the criteria and attributes under analysis can be clarified by indicating their respective

importance in the decision context, representing the preference relations of a group of decision makers as well as of each individual decision maker.

## 5. Case Study

The Remotely Piloted Aircraft Systems (RPAS), or "drones" as they are commonly known, have been providing a great technological revolution in the world, considering that these types of equipment enable a variety of applications in different areas. As discussed by Moreira et al. (2020), currently, the RPAS present the integration of high-level technologies, including sensors, radars, cameras and often firepower, enabling their use in logistics, surveillance and combat operations, providing support for tactical and strategic operations related to the defense and security of a country [118].

Specifically for military applications, RPAS have become essential devices in combat operations, increasing their demand through successful battle jobs. They provide advantages such as area sensing, tactical reconnaissance, the mitigation of human factor exposure and risk and low costs when compared to traditional aircraft applications and technologies [119].

However, if, on the one hand, military forces can exploit the capabilities enabled by an RPAS, on the other hand, the defense question must be considered in scenarios in which there is the use of remote technologies as an attack factor by opposing forces [120]. Concerning the scenario of remote air technologies, the given case is based on developing and analyzing defense strategies or anti-SARP strategies for the BN.

Regarding the development of anti-SARP strategies, first, it is necessary to obtain a better understanding of the scenario of the technologies currently present in Brazil—specifically, in the Brazilian Navy. As support in the understanding of this problematic situation, the construction of a causal map is used, which has, as the basis, the methodological concepts of the SODA from Soft OR, thus bringing a structuring of the scenario under analysis and the construction of the objectives to be achieved with a given decision making. As presented by Abuabara and Paucar-Caceres [121], there are many studies related to the applications of SODA methodology in case studies based on scenarios of strategic analysis.

The SODA methodology was proposed by [30] and is based on the study of the situation in the form of a cognitive map, seeking to reflect the points of view of each member regarding the resolution of the problem situation. This favors the interaction between those involved in the process of the decision analysis, functioning as a facilitating device for obtaining consensus among the team's actors and commitment regarding the measures that should be taken.

In order to provide the collection of data and information related to the problematic situation in question, a series of interviews seeking to provide a better understanding of the studied scenario from the point of view of the experience of people directly linked to the assessment scenario were carried out. The interviews were conducted with officers of the BN who have vast experience in national defense operations and have worked with the employment and development of remote technologies.

Concerning the group working in the problematic situation analysis, the set was composed of nine officers of BN; four of them had more than 20 years of experience in combat and defense scenarios, working as a commander in BN Ships. The other five officers had more than 10 years of experience in the technological development sector of BN, working with sensing and detecting systems for combat and defense environments.

The operated questions in the interview aimed to explore the main objectives and variables of influence in the construction of strategies for the environment against remote technologies. As an interaction model of interviews, first, individual interviews were conducted, and then we had a group interaction conversation, presenting the main objectives, feasible alternatives and actions in the construction of anti-RPAS strategies.

During the interviews, the indication of actions not necessarily related to the acquisition or development of technologies but rather to the construction of more general objectives aimed at mitigating losses and the technological delay was of great perception,

which is directly aligned with and influenced by the National Defense Strategy [3]. Other objectives, directly aligned with the issue of anti-SARP strategies, aim to maximize sensing in the coasts and territorial waters and surveillance of the Amazon area of Brazil.

In a general context, through the interactions, a set of objectives were defined, which can be divided into strategic objectives, means and ends for a given problematic situation [122]. This article will be restricted to the partial analysis of these established objectives, focusing only on objectives directly linked to the scenario of anti-SARP strategies. The causal map shown in Figure 7 presents a succinct demonstration of the constructed objectives network based on the techniques of the causal maps approach, so a given network will later be ranked based on the prioritizations of these objectives.

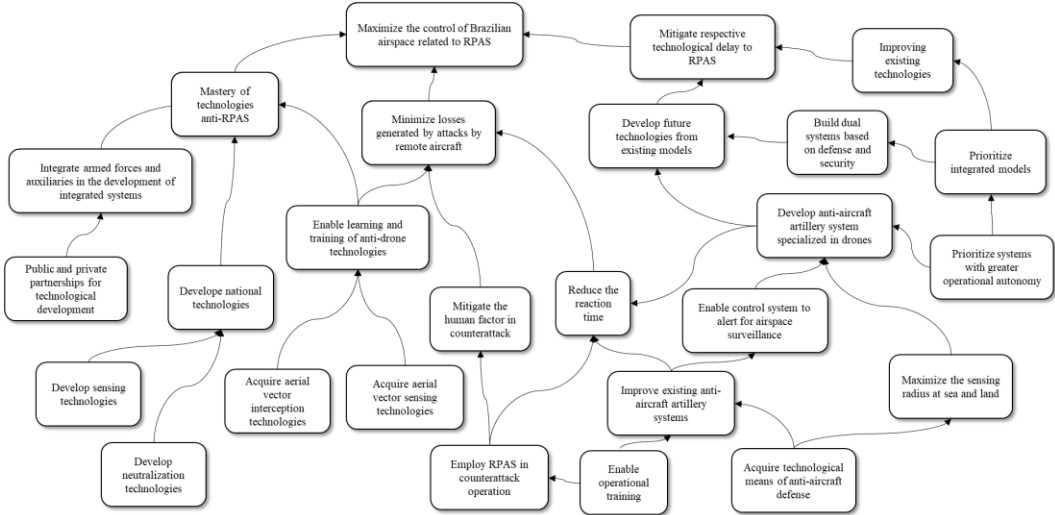

**Figure 7.** Causal map of objectives related to anti-RPAS strategies.

Regarding the maximization of the control of the Brazilian airspace related to remote technologies as the main element of the objectives network, other objectives were aligned and structured within a hierarchical network in favor of the main objective. In this context, a hierarchical structure was developed, presenting five assessment levels considering strategic, tactical and operational assessments.

Once the structure to be analyzed is defined, the SAPEVO-H$^2$ method becomes favorable for decision analysis in the constructed scenario in a way that allows, first, for the integration of multiple decision makers, clarifying their individual preferences, and, later, for the aggregation and clarification of preferences assigned to the template. Figure 8 shows the hierarchy of the elements developed; it is observed that, at the end of the structure, there are technological alternatives to be analyzed based on multiple criteria such that given alternatives can integrate the development and implementation of anti-RPAS strategies for BN.

The evaluation and numerical implementation of the SAPEVO-H$^2$ method to be carried out will be divided into two stages, firstly clarifying the preferences in the most tactical and strategic levels, characterized by the first three levels of evaluation. The analysis will be carried out at operational levels, first defining the preferences of the criteria for each element of the higher level and then analyzing the alternatives in the light of these criteria.

With a given analysis, one seeks not only to clarify the most favorable alternatives and prioritize the objectives on a global level but also to expose their respective performances at a local level, in line with the element of the higher level. In this context, the evaluation will consist of nine decision makers, one decision maker analyzing the first level of the hierarchy concerning the strategic objective, three decision makers performing the analysis of the elements of the second level, two decision makers evaluating the third level and, finally, three decision makers evaluating the operational area of the process, consisting of criteria and alternatives such that each decision maker will assess the operational part of their area of expertise. Considering that the analysis of criteria and alternatives is restricted

to an operational evaluation, this study will restrict the evaluation to sensing technologies. Figure 8 presents all the details of a hierarchical structure.

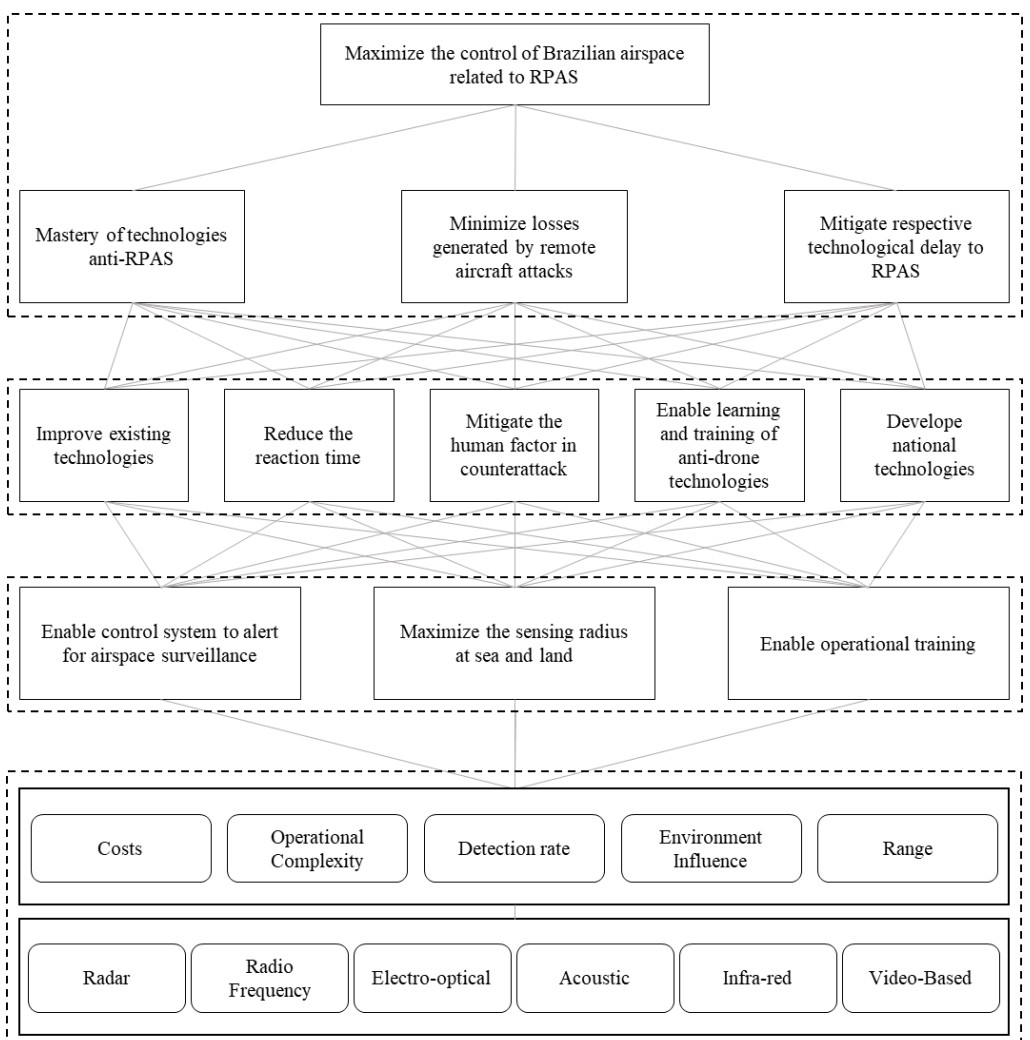

**Figure 8.** Hierarchical structure for implementing the SAPEVO-H$^2$ method.

With the given structure, the implementation of the SAPEVO-H$^2$ modeling concerns a hypothetical acquisition of sensing technologies to be implemented in the anti-RPAS strategic operations of BN. However, the given evaluation is not restricted to operational issues only; the analysis of the results takes into account all the preferences attributed at the strategic and tactical levels of the analysis. In this context, a brief description of the levels under evaluation is presented below:

- Level 1: influential objectives to the listed strategic objective;
- Level 2: tactical objectives;
- Level 3: tactical objectives;
- Level 4: evaluation criteria related to sensory technologies;
- Level 5: respective alternatives to sensory technologies.

*5.1. Numerical Implementation*

As discussed above, the assessment of the problematic situation is divided into two stages: first, evaluating the respective levels of the strategic decisions of the BN, and, later, evaluating the sets of criteria and alternatives aimed at an operational decision in the network of elements presented.

### 5.1.1. Evaluation on Strategical and Tactical Levels

The assessment of strategic levels is carried out by six decision makers (D1, D2, D3, D4, D5 and D6), where D1 assesses level 1, D2 and D3 assess level 2 and, finally, D4, D5 and D6 assess level 3 of the established network.

The first part of the evaluation concerns the analysis of the elements of the set of the first level, carried out by the decision maker D1. Because a given assessment is related to tacit information of the decision maker, the given analysis is based on the ordinal attribution scale (Table 2). In this context, the decision maker will evaluate the elements of *Mastery of technologies anti-RPAS* (MT), *Minimize losses generated by remote aircraft attacks* (ML) and *Mitigate respective technological delays to RPAS* (MD), as presented in Table 4.

**Table 4.** Evaluation of level 1 by *D1*.

|  | **MT** | **ML** | **MD** | *Punctuation* | *Normalized* | *Utility* |
|---|---|---|---|---|---|---|
| MT | 0 | 1 | −1 | 0 | 0.500 | 0.333 |
| ML | −1 | 0 | −2 | −3 | 0.250 | 0.167 |
| MD | 1 | 2 | 0 | 3 | 0.750 | 0.500 |

At the evaluation, it is necessary to carry out the analysis of level 2. At this level, five elements are considered: *Improve existing technologies* (IET), *Reduce the reaction time* (RRT), *Mitigate the human factor in a counterattack* (MHF), *Enable the learning and training of anti-drone technologies* (ELT) and *Develop national technologies* (DNT). It should be noted that a given set will be assessed concerning the three elements of the higher level previously evaluated. The assessments are shown in Tables 5–7. Table 8 presents the final result of the level 2 assessment.

**Table 5.** Evaluation of level 2 for Mastery of anti-RPAS technologies.

| | *D2* | | | | | | |
|---|---|---|---|---|---|---|---|
|  | **IET** | **RRT** | **MHF** | **ELT** | **DNT** | *Punctuation* | *Normalized* |
| IET | 0 | −1 | 3 | 1 | 0 | 3 | 0.778 |
| RRT | 1 | 0 | 3 | 2 | 1 | 7 | 1 |
| MHF | −3 | −3 | 0 | −2 | −3 | −11 | 0 |
| ELT | −1 | −2 | 2 | 0 | − | −2 | 0.500 |
| DNT | 0 | −1 | 3 | 1 | 0 | 3 | 0.778 |
| | *D3* | | | | | | |
|  | IET | RRT | MHF | ELT | DNT | *Punctuation* | *Normalized* |
| IET | 0 | 1 | 2 | 1 | 0 | 4 | 1 |
| RRT | −1 | 0 | 1 | 0 | −1 | −1 | 0.5 |
| MHF | −2 | −1 | 0 | −1 | −2 | −6 | 0 |
| ELT | −1 | 0 | 1 | 0 | −1 | −1 | 0.5 |
| DNT | 0 | 1 | 2 | 1 | 0 | 4 | 1 |
| | *D4* | | | | | | |
|  | IET | RRT | MHF | ELT | DNT | *Punctuation* | *Normalized* |
| IET | 0 | 1 | 3 | 0 | 0 | 4 | 1 |
| RRT | −1 | 0 | 2 | −1 | −1 | −1 | 0.667 |
| MHF | −3 | −2 | 0 | −3 | −3 | −11 | 0 |
| ELT | 0 | 1 | 3 | 0 | 0 | 4 | 1 |
| DNT | 0 | 1 | 3 | 0 | 0 | 4 | 1 |

In the last level corresponding to strategic contexts, three elements will be analyzed, corresponding to the five attributes of the higher level analyzed previously. Thus, the analysis variables are: *Enable the control system to alert for airspace surveillance* (ECS), *Maximize the sensing radius at sea and land* (MSR) and *Enable operational training* (EOT). In this scenario,

they consider the attributions of preferences carried out by decision makers D5 and D6, who are responsible for level 3; Tables 9–13 show this assessment in detail.

**Table 6.** Evaluation of level 2 for Minimize losses generated by remote aircraft attacks.

| | IET | RRT | MHF | ELT | DNT | *Punctuation* | *Normalized* |
|---|---|---|---|---|---|---|---|
| **D2** | | | | | | | |
| IET | 0 | −1 | −1 | 0 | 2 | 0 | 0.667 |
| RRT | 1 | 0 | 0 | 1 | 3 | 5 | 1 |
| MHF | 1 | 0 | 0 | 1 | 3 | 5 | 1 |
| ELT | 0 | −1 | −1 | 0 | 2 | 0 | 0.667 |
| DNT | −2 | −3 | −3 | −2 | 0 | −10 | 0 |
| **D3** | | | | | | | |
| IET | 0 | −2 | −1 | 0 | 3 | 0 | 0.6 |
| RRT | 2 | 0 | 1 | 2 | 3 | 8 | 1 |
| MHF | 1 | −1 | 0 | 1 | 3 | 4 | 0.8 |
| ELT | 0 | −2 | −1 | 0 | 3 | 0 | 0.6 |
| DNT | −3 | −3 | −3 | −3 | 0 | −12 | 0 |
| **D4** | | | | | | | |
| IET | 0 | 1 | 1 | 0 | 1 | 3 | 1 |
| RRT | −1 | 0 | 0 | −1 | 0 | −2 | 0 |
| MHF | −1 | 0 | 0 | −1 | 0 | −2 | 0 |
| ELT | 0 | 1 | 1 | 0 | 1 | 3 | 1 |
| DNT | −1 | 0 | 0 | −1 | 0 | −2 | 0 |

**Table 7.** Evaluation of level 2 for Mitigate respective technological delays to RPAS.

| | IET | RRT | MHF | ELT | DNT | *Punctuation* | *Normalized* |
|---|---|---|---|---|---|---|---|
| **D2** | | | | | | | |
| IET | 0 | 3 | 2 | 0 | 0 | 5 | 1 |
| RRT | −3 | 0 | −1 | −3 | −3 | −10 | 0 |
| MHF | −2 | 1 | 0 | −2 | −2 | −5 | 0.333 |
| ELT | 0 | 3 | 2 | 0 | 0 | 5 | 1 |
| DNT | 0 | 3 | 2 | 0 | 0 | 5 | 1 |
| **D3** | | | | | | | |
| IET | 0 | 2 | 2 | 1 | 1 | 6 | 1 |
| RRT | −2 | 0 | 0 | −1 | −1 | −4 | 0 |
| MHF | −2 | 0 | 0 | −1 | −1 | −4 | 0 |
| ELT | −1 | 1 | 1 | 0 | 0 | 1 | 0.5 |
| DNT | −1 | 1 | 1 | 0 | 0 | 1 | 0.5 |
| **D4** | | | | | | | |
| IET | 0 | 3 | 3 | 2 | 0 | 8 | 1 |
| RRT | −3 | 0 | 0 | −1 | −3 | −7 | 0 |
| MHF | −3 | 0 | 0 | −1 | −3 | −7 | 0 |
| ELT | −2 | 1 | 1 | 0 | −2 | −2 | 0.333 |
| DNT | 0 | 3 | 3 | 2 | 0 | 8 | 1 |

**Table 8.** Respective utilities level 2 assessments.

|  | *Mastery of Technologies* | *Minimize Losses* | *Mitigate Tech. Delay* |
|---|---|---|---|
| IET | 0.285 | 0.271 | 0.391 |
| RRT | 0.222 | 0.239 | 0.001 |
| MHF | 0.002 | 0.216 | 0.043 |
| ELT | 0.205 | 0.271 | 0.239 |
| DNT | 0.285 | 0.002 | 0.326 |

**Table 9.** Evaluation of level 3 for *Improving existing technologies*.

| | *D5* | | | | |
|---|---|---|---|---|---|
| | **ECS** | **MSR** | **EOT** | *Punctuation* | *Normalized* |
| ECS | 0 | 0 | −3 | −3 | 0 |
| MSR | 0 | 0 | −3 | −3 | 0 |
| EOT | 3 | 3 | 0 | 6 | 1 |
| | *D6* | | | | |
| | ECS | MSR | EOT | *Punctuation* | *Normalized* |
| ECS | 0 | 0 | −2 | −2 | 0 |
| MSR | 0 | 0 | −2 | −2 | 0 |
| EOT | 2 | 2 | 0 | 4 | 1 |

**Table 10.** Evaluation of level 3 for *Reducing the reaction time*.

| | *D5* | | | | |
|---|---|---|---|---|---|
| | **ECS** | **MSR** | **EOT** | *Punctuation* | *Normalized* |
| ECS | 0 | −2 | −3 | −5 | 0 |
| MSR | 2 | 0 | −1 | 1 | 0.667 |
| EOT | 3 | 1 | 0 | 4 | 1 |
| | *D6* | | | | |
| | ECS | MSR | EOT | *Punctuation* | *Normalized* |
| ECS | 0 | −3 | −3 | −6 | 0 |
| MSR | 3 | 0 | −3 | 0 | 0.5 |
| EOT | 3 | 3 | 0 | 6 | 1 |

**Table 11.** Evaluation of level 3 for *Mitigate the human factor*.

| | *D5* | | | | |
|---|---|---|---|---|---|
| | **ECS** | **MSR** | **EOT** | *Punctuation* | *Normalized* |
| ECS | 0 | 1 | −2 | −1 | 0.333 |
| MSR | −1 | 0 | −3 | −4 | 0 |
| EOT | 2 | 3 | 0 | 5 | 1 |
| | *D6* | | | | |
| | ECS | MSR | EOT | *Punctuation* | *Normalized* |
| ECS | 0 | 0 | −1 | −1 | 0 |
| MSR | 0 | 0 | −1 | −1 | 0 |
| EOT | 1 | 1 | 0 | 2 | 1 |

After evaluating the three attributes in light of the five elements of the higher level, a set of respective utilities for each context is obtained. Table 14 shows the utilities obtained in the assessment of level 3.

**Table 12.** Evaluation of level 3 for *Enable learning and training*.

| | D5 | | | | |
|---|---|---|---|---|---|
| | **ECS** | **MSR** | **EOT** | *Punctuation* | *Normalized* |
| ECS | 0 | 1 | −3 | −2 | 0.2 |
| MSR | −1 | 0 | −3 | −4 | 0 |
| EOT | 3 | 3 | 0 | 6 | 1 |
| | D6 | | | | |
| | **ECS** | **MSR** | **EOT** | *Punctuation* | *Normalized* |
| ECS | 0 | 0 | − | −3 | 0 |
| MSR | 0 | 0 | −3 | −3 | 0 |
| EOT | 3 | 3 | 0 | 6 | 1 |

**Table 13.** Evaluation of level 3 for *Developing national technologies*.

| | D5 | | | | |
|---|---|---|---|---|---|
| | **ECS** | **MSR** | **EOT** | *Punctuation* | *Normalized* |
| ECS | 0 | 1 | −1 | 0 | 0.5 |
| MSR | −1 | 0 | −2 | −3 | 0 |
| EOT | 1 | 2 | 0 | 3 | 1 |
| | D6 | | | | |
| | **ECS** | **MSR** | **EOT** | *Punctuation* | *Normalized* |
| ECS | 0 | 1 | −2 | −1 | 0.333 |
| MSR | −1 | 0 | −3 | −4 | 0 |
| EOT | 2 | 3 | 0 | 5 | 1 |

**Table 14.** Respective utilities level 3 assessments.

| | *Improve Exiting Technologies* | *Reduce Reaction Time* | *Mitigate Human Factor* | *Enable Learning and Training* | *Develop National Technologies* |
|---|---|---|---|---|---|
| ECS | 0.010 | 0.004 | 0.143 | 0.091 | 0.293 |
| MSR | 0.010 | 0.367 | 0.001 | 0.001 | 0.003 |
| EOT | 0.980 | 0.629 | 0.856 | 0.908 | 0.704 |

Once it has been completed, the analysis of the three strategic levels obtains the respective utilities of its elements, and the evaluation ends at this stage. In the next section, the analysis of the operational levels of the problem is carried out. Afterwards, the analysis of the local and global results obtained is presented.

5.1.2. Evaluation of Operational Levels

In the context of the development of systems integrated with anti-RPAS strategies, it should be noted that a given assessment area is divided into three types of technologies, as shown in Figure 9: Detection, Classification and Neutralization [123]. For the purposes of the study and the methodological exploration, the analysis at operational levels is restricted to analyzing detection technologies.

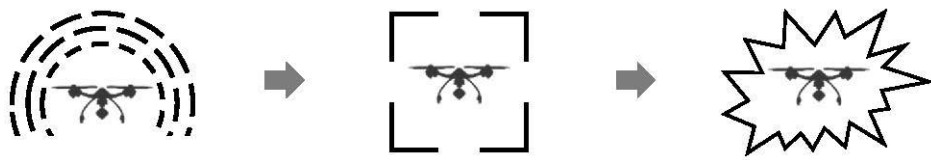

**Figure 9.** Anti-RPAS systematic based on detection, classification and neutralization.

In this scenario, we assessed five criteria as the most influential in operational matters. Four are analyzed under a qualitative perspective, and one is evaluated with a quantitative methodological basis, as presented in Section 4.2. Below is a description of the criteria under evaluation.

- Costs: Variable related to the expenditure of resources destined for the acquisition and maintenance of the technology; for the study and data preservation, the variable will be analyzed qualitatively;
- Operational Complexity: Given criteria are aligned with the degree of complexity present in operation and implementation of each type of technology, evaluated qualitatively;
- Detection Rate: The criterion in question represents how favorable a technology is for detecting RPAS when it is applied in different types of scenarios, considering targets of different sizes;
- Environment Influence: Considering external variables such as climate, movement and ambient light, for example, a given variable represents the adaptation and operation relationship of a given technology in unstable environment scenarios or with restrictions, this being analyzed qualitatively;
- Range: Given variable quantitatively represents the detection range limit of each technology.

It is noteworthy that, for a given analysis, three individual assessments of the set of criteria are carried out; these concern the elements of the upper level (*Enable the control system to alert for airspace surveillance, Maximize the sensing radius at sea and land and Enable operational training*). Therefore, decision makers D7, D8 and D9 will evaluate the criteria based on their respective areas of expertise, namely, *Airspace Surveillance*, *Maritime and Land Sensing* and *Operational Studies*.

As for the alternatives, six types of technologies were listed, which were evaluated in light of the five criteria. A description of the alternatives is presented.

- Radar: the technology is related to the emission of radio waves, reflected by the possible targets to be detected;
- Radio Frequency: Model operated based on RPAS communication with ground control centers, carried out by employing radio wave transmission;
- Electro-optical: Sensory technologies that are operable through the conversion of light rays into electronic signals, enabling the detection and identification of aerial objects;
- Acoustic: The technology in question operates based on the detection of acoustic signatures emitted by different types of drones, using a database for comparisons and the definition of the detected object;
- Infrared: Model designed to operate based on the heat emitted by an RPAS, detecting and analyzing the thermal requirements of the processed image;
- Video-Based: Technologies based on the integration of cameras and sensors aimed at identifying objects moving in the airspace.

Once the variables to be analyzed are defined, the criteria are first evaluated based on the experience and preference of each decision maker in their area of expertise. Tables 15–17 present the three analyses with their respective scores and weights of the criteria in each scenario.

Once the criteria have been evaluated, it is necessary to evaluate the set of alternatives in light of them. In this way, each of the three decision makers will determine their preferences for the variables based on multiple criteria.

Considering four qualitative criteria and one quantitative criterion, first, the evaluation of the quantitative criterion is carried out, related to the range of reach of each of the six technologies. Table 18 presents the approximate quantitative performances of the alternatives in the range criterion, and Table 7 also shows the function established for the evaluation process along with their respective thresholds, defined by decision makers D7, D8 and D9.

**Table 15.** Evaluation of criteria on level 4 by D7 for *Airspace Surveillance*.

| | Costs | Operational Complexity | Detection Rate | Environment Influence | Range | *Punctuation* | *Weights* |
|---|---|---|---|---|---|---|---|
| | | | *D7* | | | | |
| | | | *Min Sum = −12 and Max Sum = 12* | | | | |
| Costs | 0 | −1 | −2 | −3 | −1 | −7 | 0.0833 |
| Operational Complexity | 1 | 0 | −1 | −2 | 0 | −2 | 0.1667 |
| Detection Rate | 2 | 1 | 0 | −1 | 1 | 3 | 0.2500 |
| Environment Influence | 3 | 2 | 1 | 0 | 2 | 8 | 0.3333 |
| Range | 1 | 0 | −1 | −2 | 0 | −2 | 0.1667 |

**Table 16.** Evaluation of criteria on level 4 by D8 for *Maritime and Land Sensing*.

| | Costs | Operational Complexity | Detection Rate | Environment Influence | Range | *Punctuation* | *Weights* |
|---|---|---|---|---|---|---|---|
| | | | *D8* | | | | |
| | | | *Min Sum = −12 and Max Sum = 12* | | | | |
| Costs | 0 | 0 | −1 | −1 | −3 | −5 | 0.1167 |
| Operational Complexity | 0 | 0 | −1 | −1 | −3 | −5 | 0.1167 |
| Detection Rate | 1 | 1 | 0 | 0 | −2 | 0 | 0.2000 |
| Environment Influence | 1 | 1 | 0 | 0 | −2 | 0 | 0.2000 |
| Range | 3 | 3 | 2 | 2 | 0 | 10 | 0.3667 |

**Table 17.** Evaluation of criteria on level 4 by D8 for *Operational studies*.

| | Costs | Operational Complexity | Detection Rate | Environment Influence | Range | *Punctuation* | *Weights* |
|---|---|---|---|---|---|---|---|
| | | | *D9* | | | | |
| | | | *Min Sum = −12 and Max Sum = 12* | | | | |
| Costs | 0 | −2 | 1 | 2 | 0 | 1 | 0.1857 |
| Operational Complexity | 2 | 0 | 3 | 3 | 2 | 10 | 0.3143 |
| Detection Rate | 1 | −1 | 2 | 3 | 1 | 6 | 0.2571 |
| Environment Influence | −2 | −3 | −1 | 0 | −2 | −8 | 0.0571 |
| Range | 0 | −2 | 1 | 2 | 0 | 1 | 0.1857 |

**Table 18.** Performance of alternatives in the range criterion and thresholds established by D7, D8 and D9.

| Alternatives | Range | Decision Maker | Preference Function | $L_{min}$ | $L_{max}$ |
|---|---|---|---|---|---|
| Radar | 1300 ft | D7 | Linear | 200 ft | 1000 ft |
| Radio Frequency | 1400 ft | D8 | Linear | 250 ft | 1200 ft |
| Electro-optic | 450 ft | D9 | Linear | 50 ft | 800 ft |
| Acoustic | 150 ft | | | | |
| Infrared | 900 ft | | | | |
| Video-based | 350 ft | | | | |

Based on the processing of the preferences established among the alternatives under the criteria, a set of scores ranging from 0 to 1 were obtained for each criterion, with 0 being unfavorable and 1 being strictly favorable, as shown in Tables 19–21.

**Table 19.** Preferences of the alternatives in the criteria according to the evaluation of D7.

| Alternatives | Costs | Operational Complexity | Detection Rate | Environment Influence | Range |
|---|---|---|---|---|---|
| Radar | 0 | 0.500 | 0.773 | 1 | 1 |
| Radio Frequency | 0 | 0 | 1 | 1 | 1 |
| Electro-optic | 0.333 | 0 | 0.500 | 0.667 | 0.313 |
| Acoustic | 1 | 1 | 0 | 0 | 0 |
| Infrared | 0.667 | 0.500 | 0.500 | 0.333 | 0.875 |
| Video-based | 0.667 | 0 | 0.227 | 0 | 0.188 |

**Table 20.** Preferences of the alternatives in the criteria according to the evaluation of D8.

| Alternatives | Costs | Operational Complexity | Detection Rate | Environment Influence | Range |
|---|---|---|---|---|---|
| Radar | 0 | 1 | 1 | 0.810 | 1 |
| Radio Frequency | 0 | 1 | 1 | 1 | 1 |
| Electro-optic | 0.500 | 0 | 0.667 | 0.238 | 0.211 |
| Acoustic | 1 | 0.500 | 0 | 0 | 0 |
| Infrared | 0.500 | 1 | 0.333 | 0.524 | 0.684 |
| Video-based | 1 | 0.500 | 0 | 0 | 0.105 |

**Table 21.** Preferences of the alternatives in the criteria according to the evaluation of D9.

| Alternatives | Costs | Operational Complexity | Detection Rate | Environment Influence | Range |
|---|---|---|---|---|---|
| Radar | 0.150 | 0 | 0.810 | 1 | 1 |
| Radio Frequency | 0 | 0.500 | 1 | 1 | 1 |
| Electro-optic | 1 | 0.500 | 0.238 | 0.500 | 0.533 |
| Acoustic | 1 | 1 | 0 | 0 | 0.133 |
| Infrared | 0.750 | 0 | 0.524 | 0.500 | 1 |
| Video-based | 1 | 0 | 0 | 0 | 0.400 |

Concerning the preferences established in each evaluated criterion, it is possible to carry out the aggregation process in such a way that it provides for establishing the performance of the set of alternatives within each analyzed level and for each element belonging to each level. In this scenario, the next section (Section 5.1.3) performs a succinct analysis of the aggregated utilities of the alternatives for each level built.

5.1.3. Aggregation Analysis of Alternatives in Each Level

Carrying out the process of the additive aggregation of the alternatives following the weights obtained in the criteria in each of the three areas of analysis, a set of utilities representing a relative degree of preference of each alternative within each evaluated scenario was provided. In this context, Table 22 presents the list of utilities established, respectively, for the level 3 variables.

Since it is a model based on a hierarchical structure, it is possible to recognize the performance of alternatives at all established levels. In this scenario, Tables 23 and 24 present the performance of alternatives at levels 2 and 1, respectively.

By completing the evaluation of the alternatives in a global way, that is, taking into account all the elements of the hierarchy, in the end, it is possible to obtain the ordering of the detection technologies because of the strategic objective, presenting the order of preferences of the technologies to the evaluated context. It is noteworthy that, as with the alternatives, a given model provides an extension of the aggregation assessment to all elements belonging to the hierarchical network, making it possible to clarify their

respective degrees of importance in the various types of objectives or established strategic requirements. Table 25 presents the global performance of alternatives along with their respective ranking positions.

**Table 22.** Alternative performance on level 3.

| Alternatives | *Airspace Surveillance* | *Maritime and Land Sensing* | *Operational Studies* |
| --- | --- | --- | --- |
| Radar | 0.777 | 0.845 | 0.479 |
| Radio Frequency | 0.750 | 0.883 | 0.657 |
| Electro-optic | 0.427 | 0.316 | 0.532 |
| Acoustic | 0.250 | 0.175 | 0.525 |
| Infrared | 0.521 | 0.597 | 0.488 |
| Video-based | 0.144 | 0.214 | 0.260 |

**Table 23.** Alternative performance on level 2.

| Alternatives | *Improve Ex. Tech.* | *Reduce React. Time* | *Mitigate Human Fac.* | *Enable Learn. Train.* | *Develop Nat. Tech.* |
| --- | --- | --- | --- | --- | --- |
| Radar | 0.485 | 0.614 | 0.522 | 0.506 | 0.567 |
| Radio Frequency | 0.660 | 0.741 | 0.671 | 0.666 | 0.685 |
| Electro-optic | 0.529 | 0.452 | 0.516 | 0.522 | 0.500 |
| Acoustic | 0.519 | 0.395 | 0.485 | 0.499 | 0.443 |
| Infrared | 0.490 | 0.528 | 0.493 | 0.491 | 0.498 |
| Video-based | 0.258 | 0.243 | 0.243 | 0.249 | 0.226 |

**Table 24.** Alternative performance on level 1.

| Alternatives | *Mastery of Anti-RPAS Tech.* | *Minimize Losses by Attack* | *Mitigate Respective Tech. Delay* |
| --- | --- | --- | --- |
| Radar | 0.542 | 0.530 | 0.519 |
| Radio Frequency | 0.686 | 0.683 | 0.670 |
| Electro-optic | 0.502 | 0.506 | 0.517 |
| Acoustic | 0.466 | 0.477 | 0.488 |
| Infrared | 0.501 | 0.500 | 0.493 |
| Video-based | 0.244 | 0.249 | 0.245 |

**Table 25.** Global performance and a final ranking of alternatives.

| Alternatives | **Global Score** | *Ranking* |
| --- | --- | --- |
| Radar | 0.528 | 2° |
| Radio Frequency | 0.678 | 1° |
| Electro-optic | 0.510 | 3° |
| Acoustic | 0.479 | 5° |
| Infrared | 0.497 | 4° |
| Video-based | 0.245 | 6° |

Considering the obtained results, we observe that the Radio Frequency technology, even with a relatively high operating cost, presents itself as the most favorable globally and in the aggregation phases, along with the hierarchy levels. The Radar and Electro-optic alternatives were also favorable during the analyses as a second or third implementation option. As crucial as clarifying the most promising alternatives, it is necessary to highlight that the video-based alternative was not profitable for implementation in most of the evaluations; this is due to the high need for climatic stability and the low detection range, which is only viable for study purposes or for the integration with other detection systems.

5.1.4. Outranking Analysis

An outranking analysis is carried out at the three levels of strategic evaluation to enrich the assessment. The purpose of each assessment is to present an overview of the

performance of the variables at each level, thus clarifying the most favorable or preferred variables at each level.

Thus, levels 1, 2 and 3 generate a set of matrices representing the local performance between the variable, along with the outranking index obtained for each level. A graphical analysis of processed preferences is also made possible for a given evaluation. First, we analyze the elements of level 3: Enable the control system to alert for airspace surveillance (ECS), Maximize the sensing radius at sea and land (MSR) and Enable operational training (EOT). In this way, Table 26 and Figure 10 present the outranking relations between the elements on level 3.

**Table 26.** Outranking relation between the elements on level 3.

|  | ECS | MSR | EOT | Outranking Index |
|---|---|---|---|---|
| ECS | 0 | 0.007417 | −0.1488 | −0.141 |
| MSR | −0.00742 | 0 | −0.15622 | −0.164 |
| EOT | 0.148803 | 0.15622 | 0 | 0.305 |

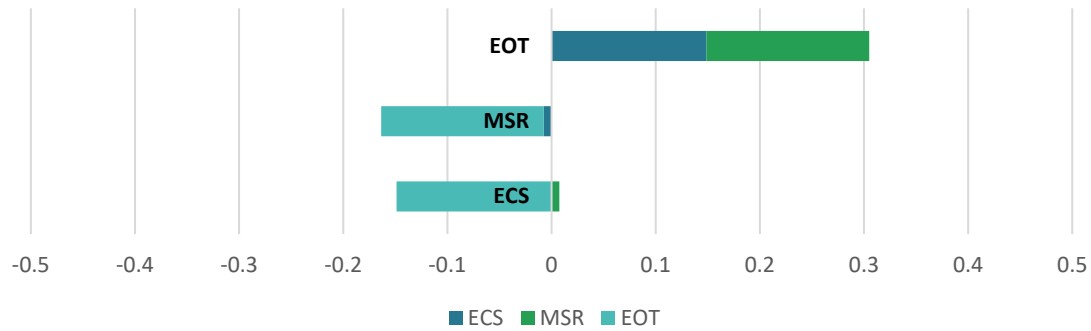

**Figure 10.** Outranking graphic regarding level 3.

By analyzing the graphic in Figure 10, we observe that the EOT element had a greater preference among the level 3 variables, exposing a strict outranking performance. It is understood that investment in technology and the expansion of operational issues are not enough; thus, there is a need for prioritizing effective operational training to ensure the effectiveness of anti-RPAS strategies.

Level 2 is evaluated based on its five elements, which are: Improve existing technologies (IET), Reduce the reaction time (R RT), Mitigate the human factor in a counterattack (MHF), Enable the learning and training of anti-drone technologies (ELT) and Develop national technologies (DNT). Table 27 and Figure 11 show the outranking relation between the variables.

**Table 27.** Outranking relation between the elements on level 2.

|  | IET | RRT | MHF | ELT | DNT | *Outranking Index* |
|---|---|---|---|---|---|---|
| IET | 0 | 0.194239 | 0.259671 | 0.093621 | 0.069753 | 0.617284 |
| RRT | −0.19424 | 0 | 0.065432 | −0.10062 | −0.12449 | −0.35391 |
| MHF | −0.25967 | −0.06543 | 0 | −0.16605 | −0.18992 | −0.68107 |
| ELT | −0.09362 | 0.100617 | 0.166049 | 0 | −0.02387 | 0.149177 |
| DNT | −0.06975 | 0.124486 | 0.189918 | 0.023868 | 0 | 0.268519 |

As shown in Figure 11, the ELT element outranked the other elements of level 2, also presenting a good distance between the DNT element, in second place. Given the graphic model, in addition to making it possible to understand the element with the highest score, they also expose how much each alternative overclassified the others within the amount, which highlights the low preference performance of the MHF element, which is totally outranked.

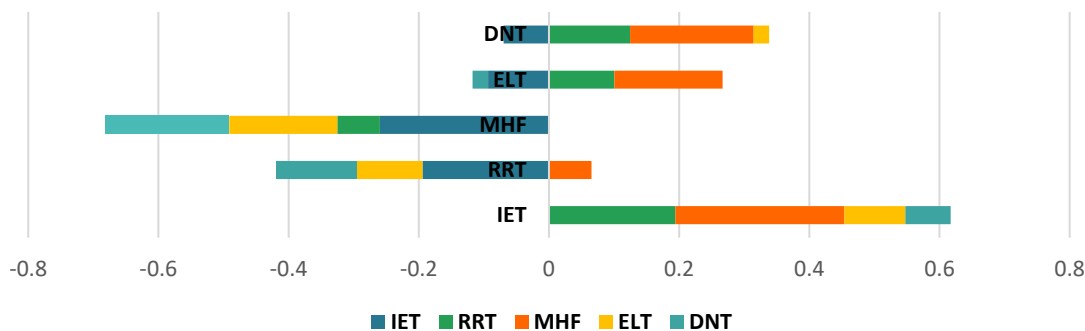

**Figure 11.** Outranking graphic regarding level 2.

At the end of the given evaluation, the outranking analysis for the elements of the first level is carried out: Mastery of anti-RPAS technologies (MT), Minimize losses generated by remote aircraft attacks (ML) and Mitigate respective technological delays to RPAS (MD). In this context, the three elements are evaluated, with the results shown in Table 28 and Figure 12.

**Table 28.** Outranking relation between the elements on level 1.

|  | **MT** | **ML** | **MD** | *Outranking Index* |
|---|---|---|---|---|
| MT | 0 | 0.167 | −0.167 | 0 |
| ML | −0.167 | 0 | −0.333 | −0.5 |
| MD | 0.167 | 0.333 | 0 | 0.5 |

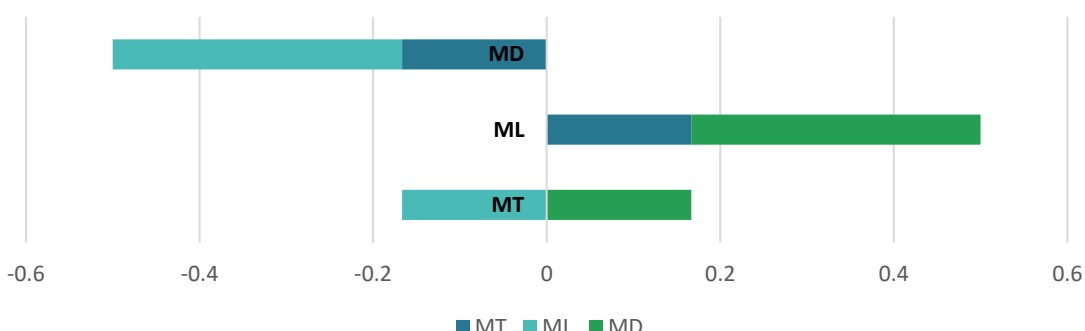

**Figure 12.** Outranking graphic regarding level 1.

Concluding the evaluation, with the results shown in Table 28 and Figure 12, it is understood that the focus of BN on the anti-RPAS strategic development scenario is not intended to be a reference in the development in the world, but the priority concern is to mitigate the losses generated by remote attacks, which can be financial, material, social or human losses, by the improvement of technological backwardness.

## 6. Discussion

The models based on the concepts of the MCDA require a set of techniques that provide the evaluation of real-world problems, which are composed of uncertainties, risks, subjectivity and divergent opinions, for example [124]. Contextualizing the presented study, the proposed model, through its axiomatic structure and established logical process, enabled the treatment and evaluation of subjective and deterministic information, respective to the preferences of the decision makers at each level and the decision scenario analyzed within the established hierarchy of evaluations.

One of the main points of gain related to the approach presented reflects the feasibility of integrating multiple decision makers into a decision-making process, distributing

them according to their areas and contexts of knowledge, performance and expertise. It is noteworthy that, given the application of the proposed model, it is guaranteed that each decision maker or evaluator of the decision-making process will be operating and evaluating scenarios and variables that have the domain, thus enabling the mitigation of the risk of submitting a decision-making person in an evaluation of variables of which the necessary knowledge is not available in the final decision process [125]. In addition to the modeling explored, the feasibility of a given decision, with multiple levels of evaluation, was also exposed to the analysis with multiple decision makers and a single decision maker in an aggregation structure.

In a decision-making analysis, the set of utilities obtained in the variables of the decision context has the main influence on the suggestion of the final decision. In this context of the proposed model, the peer-to-peer evaluation is efficient, considering that it is not always simple to perform a global preference assignment of a variable as a whole [126]; thus, the clarification of global preferences is provided through a sequence of preference assignments between only two variables. Additionally, in the given model that was still proposed in its axiomatic structure, the feasibility of testing the consistencies of the attributions performed in the evaluations clarified the consistency relationships in the sets of preferences designated in the assessments. As a limitation of the model, we note that the model still does not present a treatment for uncertainty; in this way, we intend to work on this gap in future studies for a greater robustness of the model.

In a complementary way, based partially on the approach presented in [127], some limitations and comparisons are necessary, thus exposing some general characteristics of the viability of the proposed model compared to the traditional models already existing in the literature. One of the major limitations identified as the motivation for the developed method concerns the consideration of only two levels of evaluation, established as criteria and alternatives; as already demonstrated in this study, in many real cases, it is necessary to include other levels, thus transcribing the necessary objectives to reach the final decision making. In this way, Table 29 exposes the main characteristics of the proposed model in relation to the main models present in the literature, listing the limitations, feasibility and respective particularities of all the models discussed.

**Table 29.** Classical models comparison and characteristics.

| Characteristics | SAPEVO-H$^2$ | SAPEVO-M | AHP | PROMETHEE | ELECTRE |
|---|---|---|---|---|---|
| Problematic | Ranking | Ranking | Ranking | Ranking | Clustering |
| Limit of Levels | Without limits of levels (objectives, criteria and alternatives). | Two levels (criteria and alternatives). | Two levels in the traditional modeling (criteria and alternatives). | Two levels (criteria and alternatives). | Two levels (criteria and alternatives). |
| Data Nature | Qualitative and quantitative | Qualitative | Qualitative | Quantitative | Quantitative |
| Treatment For Group Analysis | Enable a group analysis, where all decision makers can evaluate all the elements of the hierarchy, or, if necessary, the evaluation is related just to the variables of expertise. | All groups need to evaluate all variables. | Performs individual assessments, and then applies the geometric mean as the aggregation. | Use the variation PROMETHEE-GDSS considering the mean of the results. | Use extensions to enable group analysis. |
| General Limitations | All elements and levels need to be evaluated.<br><br>If there are a large number of elements in just one level, it is indicated to work with the cardinal input instead of the comparison analysis. | All data need to be evaluated as the ordinal input.<br><br>Do not present a consistency analysis concerning pairwise evaluations. | In the case of many elements, the number of comparisons becomes exhaustive and non-trivial.<br><br>Uses just the Saaty scale as a model of input. | Does not consider qualitative or subjective information treatment.<br><br>The weights need to be established by the cardinal input. | Does not offer subjective treatment for the inputs.<br><br>The weights need to be established as the cardinal input. |

Respectively, through the case study approach, it was possible to understand and partially structure the problem related to the construction of anti-RPAS strategies, transcribing objectives to be achieved through the preferences assigned among them at their respective levels. The implementation of SAPEVO-H$^2$ modeling initially allowed for the clarification of the variables in their evaluation levels, thus enabling the aggregation between them at different levels and presenting the aggregate performance of the alternatives, criteria, objectives and elements that make up the structured hierarchy of the analyzed case, exposing the ranking of these to different scenarios. Regarding the consistency of the peer-to-peer evaluations performed, high and medium consistency rates were found in all analyzed parts. In addition to the aggregation model, the outranking analysis was also of great value when it made it possible to clarify the distance between the performances obtained at the end of the evaluation.

As for the detection technologies explored, it is worth mentioning that it is also valid to evaluate these devices in an integrated way so that a given technology can prove more favorable for integration with others, thus providing the construction of a kind of synergy so that the return of the application of two or more technologies in an integrated way will be greater than the sum of these operating individually. It is also indicated for future work that the analysis and integration of machine learning technologies to the analyzed devices and learning algorithms can also be evaluated based on the analysis of multiple variables.

We should emphasize that all preferences established in the study are based on the perspectives and vision of officers that worked on the analysis of the case, presenting the vision of some military organizations of the BN and no exposure to the vision of all of the BN or other Brazilian military forces, defense departments or remote technologies companies. Aligned with some limitations of the study, we emphasize that the study brings the perception of decision making that is restricted for constructions and the evaluation of anti-RPAS strategies in the sensing environment for defense, where the change in decision makers or the political environment can provide the establishment of different variables, objectives and alternatives, exposing a new hierarchical structure with a variety of new actions as favorable solutions.

In addition to the discussion, we identify the need to construct a computational model, given the feasibility of a computational tool that is necessary for the expansion and applicability of a mathematical model [124]. For a trivial implementation of the proposed model, it will be sought that the computational platform to be developed operates online, enabling the integration and distribution of decision makers in the given evaluation scenario and then allowing for the exploration of numerical and graphical resources in the aggregation, exposure and discussion of results.

## 7. Conclusions

This study aimed to propose a methodological approach based on the concepts of the MCDA, named SAPEVO-H$^2$. The given modeling presents a new approach based on the methodology of SAPEVO methods, operating under the perspective of a multilevel hierarchical evaluation structure and integrating multiple decision makers in an evaluation format capable of treating and evaluating data of a quantitative and qualitative nature simultaneously.

The model, structured in a set of assessment steps, performs the integration of quantitative assessment, through cardinal inputs, in the ordinal assessment model, which is present in the SAPEVO methodologies. Related to the dynamics of implementation, the given model provided the construction of a group of decision makers, which partitioned the assessments of the elements of their expertise, which are divided into different levels of impact on the problematic situation, based on strategic, tactical and operational areas. As a test of the consistency of the preferences inputted in the pairwise evaluations, analysis modeling was proposed in the axiomatic structuring, bringing more robustness and consistency to the decision-making analysis. As for the study of the results, a given model makes it possible to explore the performance of the elements present in the hierarchy at all levels

of assessment, operating on an additive aggregation approach, and, additionally, it also enables an outranking analysis based on the outranking relation of the elements belonging to each level assessed.

Concerning the case study addressed, a real problematic situation was presented, which was related to the evaluation of anti-RPAS strategies for BN, first exposing an understanding of the problem and its respective partial structure using the causal maps approach, thus enabling the implementation of the SAPEVO-H$^2$ method. Through this analysis, it was possible to clarify the preferences among the elements belonging to each level evaluated, showing the respective influences at the operational level and, later, the performance of the technologies at the operational level in all elements of the strategic levels. Additionally, an outranking evaluation related to the tactical and strategic elements of the problematic situation was explored, clarifying their respective degrees of superiority over the other elements belonging to each level analyzed.

In this context, we concluded that the given model has the potential to be applied in several areas of study through its flexibility and adaptability of the variables in a high-level decision-making context. For future studies, we established the evolution of this study concerning the construction of other objectives and alternatives aligned with the sensing environment for anti-RPAS strategies in defense; on the other hand, we search for the application of the proposed model in other case studies, looking to clarify points of improvement in the model and increase its robustness. Additionally, the development of a computational platform for online access will be worked on, integrating several decision makers in a dynamic evaluation environment, operating in real-time, exploring numerical and graphical resources and thus supporting better decision making.

**Author Contributions:** Conceptualization, M.Â.L.M. and I.P.d.A.C.; methodology, M.Â.L.M. and I.P.d.A.C.; software, M.Â.L.M.; validation, M.Â.L.M., F.C.A.S. and I.P.d.A.C.; formal analysis, M.Â.L.M. and F.C.A.S. investigation, M.Â.L.M.; resources, F.C.A.S.; data curation, C.F.S.G. and M.d.S.; project administration, C.F.S.G. and M.d.S.; funding acquisition, F.C.A.S., C.F.S.G. and M.d.S. All authors have read and agreed to the published version of the manuscript.

**Funding:** This research received no external funding.

**Institutional Review Board Statement:** Not Applicable.

**Informed Consent Statement:** Not Applicable.

**Data Availability Statement:** Not Applicable.

**Conflicts of Interest:** The authors declare no conflict of interest.

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
