# Peer review of "SAPEVO-H² a Multi-Criteria Systematic Based on a Hierarchical Structure: Decision-Making Analysis for Assessing Anti-RPAS Strategies in Sensing Environments"

_processes, doi:10.3390/pr11020352_

Round 1

Reviewer 1 Report

The subject of the paper is interesting and in line with the aims and scope of the Journal. The paper provides useful information and interesting results. However, the paper is badly structured. It is too partitioned, and without numeration, therefore very hard to follow. Also, some main elements of the scientific paper are missing. The authors did not identify the aim, focus, limitations, implications, and scientific contributions of the paper. There is also a question about the originality and novelty of the paper. More detailed comments are provided below.

1.     The abstract is not written well:

a)     Main aim and focus of the paper are not clear. Is the aim of the paper to develop a new decision-making methodology or to solve the problem of assessing anti-RPAS strategies?

b)    About the previous, once it is clear what is the main focus of the paper, the appropriate background should be provided. The authors must explain why is this problem worth investigating and what they expect to achieve in their study.

c)     Main results, conclusions, and scientific contributions are not highlighted in the abstract.

2.     The introduction also lacks a clear identification of the aim, main results, conclusions, and contributions of the paper.

3.     The authors did not identify the research gaps (based on the reviewed literature) that their paper is trying to cover.

4.     It is unclear what is novel in the proposed methodology. The used methods are already known and applied in the literature. The authors should highlight what if anything is novel. I am also concerned about the level of already published materials from this paper since at least two papers are addressing the same problem and using the same methodology („Moreira, M. Â. L., dos Santos, M., & Gomes, C. F. S. Seleção de um Sistema de Aeronave Remotamente Pilotada (SARP) para a Marinha do Brasil: um olhar sob a perspectiva do método SAPEVO-H².“ and „Moreira, M. Â. L., Gomes, C. F. S., & dos Santos, M. SAPEVO-H² Abordagem Multicritério Baseada em Estrutura Hierárquica: Análise de SARP para a Marinha do Brasil.“).

5.     More information should be provided regarding the case study. The authors should explain in more detail how were alternatives established and by whom, how are criteria established, who were the decision-makers, and what is their background (experience), etc.

6.     The authors should consider placing some of the tables in the appendix or Supplementary material.

7.     Discussion should be supplemented with the limitations of the study, as well as with some theoretical and practical (managerial) implications (who and for what can use the results of the study).

8.     Again, from the future research directions it is not clear what was the main focus of the paper. The authors should provide more general future research directions, not only about the methodology but also about the problem. Future research directions should be interesting to most of the Journal readership.

9.     Extensive self-citation is noted in the paper. Is it truly necessary to cite all those papers authored by the authors of this paper?

10.  There are certain technical issues:

a)     The abstract slightly exceeds the limited number of words, as indicated in the instructions for authors.

b)    The authors did not fully use the provided template for formatting the paper.

c)     Sections and sub-sections are not numbered (some of them at all and some incorrectly).

d)    Figure titles should be placed below the figure, not above.

e)     Acronyms/Abbreviations/Initialisms should be defined the first time they appear in each of three sections: the abstract; the main text; the first figure or table. For example, abbreviations „SAPEVO-M“, „SODA“, etc. are not defined in the abstract. Check the rest of the paper.

f)     References should be quoted in the text using the numbers in square brackets in order of their appearance.

g)    All references quoted in the text must be present in the reference list and vice versa. For example, the reference „Van Hoazn and Ha (2021)“ is not listed in the reference list. Check the rest of the paper.

h)    Figure 3 is almost completely unreadable.

Author Response

PUNCTUAL RESPONSE TO THE REVIEWER 1

Rio de Janeiro, January 12, 2023.

Dear Reviewer of the Processes Journal,

First, we would like to thank you for all attention and contribution for the study. In this way, we describe below the treatment of the main changes proposed by the reviewers for the manuscript "SAPEVO-H² a Multi-Criteria Systematic Based on Hierarchical Structure: Decision-Making Analysis for Assessing anti-RPAS Strategies in Sensing Environment".

Reviewer#1

The abstract is not written well.

R.

The abstract has been updated. (Lines 11 – 25)

“Regarding high-level and complex decision-making scenarios, the study presents an extensive approach to the Simple Aggregation of Preferences Expressed by Ordinal Vectors - Multi Decision Making method (SAPEVO-M). In this context, the modeling proposal, named SAPEVO – Hybrid and Hierarchical (SAPEVO-H²), the objective of this study, based on the concepts of multi-criteria analysis, provides the evaluation of alternatives under the light of multiple criteria and perceptions, enabling the integration of the objectives of a problem, transcribed into attributes, and structured in a hierarchical model, analyzing qualitative and quantitative data through ordinal and cardinal entries, respectively. As a case study is carried out a decision analysis concerning the defense strategies against anti-Remotely Piloted Aircraft Systems (RPAS) strategies for the Brazilian Navy. Using techniques of causal maps approach based on Strategic Options Development and Analysis (SODA) methodology, the problematic situation is structured for numerical implementation, demonstrating the performance of objectives and elements of hierarchical structure. As result, are presented rankings concerning objectives and anti-RPAS technologies, based on the treatment of subjective information. In the end, is discussed the main contribution of the study and its limitations, along with the conclusions and some proposals for future studies.”

The introduction also lacks a clear identification of the aim, main results, conclusions, and contributions of the paper.

R.

The introduction section has been updated. (Lines 107 – 118)

“In search to provide the feasibility of the approach proposal, a case study concerning the analysis of anti-RPAS strategies is explored. For a better understanding of the problematic situation, is used techniques of causal maps based on the principles of Strategic Options Development and Analysis (SODA) [26], enabling the clarification of objectives intrinsic to the analysis of strategies, structuring these variables for their treatment and evaluation through the methodological proposal to be presented, performing the integration of multiple decision-makers at different hierarchical levels and knowledge of the variables belonging to the problematic situation.

As the main contribution of this study, we search to provide the problematic situation structuring regarding some of the main objectives of anti-RPAS strategies for the Brazilian Navy, clarifying the priorities between the elements into evaluation, through outranking analysis by the performance of alternatives, serving as an aid in the decision-making.”

The authors did not identify the research gaps (based on the reviewed literature) that their paper is trying to cover.

R.

The introduction section has been updated. (Lines 81 – 89)

“As presented in [19], in real problematic situations of decision analysis, uncertainty variables are intrinsic to the assessment [20], especially in environments involving multiple decision-makers belonging to different levels of strategic, where members generally disagree about the parameter values, and preference assignments [21]. Nevertheless, even though group decisions are expected in analyzing real situations, few models consider the formation of subgroups to deal with problems at strategic, tactical, and operational levels in particular [22], in other words, most of the models do not consider hierarchical analysis, providing the integration of different levels of expertise in the decision-making process.”

It is unclear what is novel in the proposed methodology. The used methods are already known and applied in the literature. The authors should highlight what if anything is novel.

R.

The previous studies were just initial versions of the proposal, limited only to the analysis of drones. In this article, we present the complete and final version of the mathematical modelling, integrated consistency validation and overclassification analysis, presenting not only the ranking of alternatives, but the distances between the scores obtained. As for the case study, a new decision-making environment is approached, analyzing objectives and technologies for building strategies to combat remote technologies against the national defense environment.

More information should be provided regarding the case study. The authors should explain in more detail how were alternatives established and by whom, how are criteria established, who were the decision-makers, and what is their background (experience), etc.

R.

The case study has been updated. (Lines 513 – 528)

“In search to provide the collection of data and information related to the problematic situation in question, were carried out a series of interviews seeking to provide a better understanding of the studied scenario from the point of view of the experience of people directly linked to the assessment scenario. The interviews were conducted with officers of the BN who have vast experience in national defense operations, in particular, having worked with the employment and development of remote technologies.

Concerning the group working in the problematic situation analysis, the set was composed of nine officers of BN, four of them had more than 20 years of experience in combating and defense scenarios, working as a commander in BN Ships. The other five officers, had more than 10 years of expertise in the technological development sector of BN, working with sensing and detecting systems for combating and defense environment.

The operated questions in the interview aimed to explore the main objectives and variables of influence in the construction of strategies for the environment against remote technologies. As an interaction model of interviews, first conducted individual interviews, and then, we had a group interaction conversation, presenting the main objectives, and feasible alternatives ad actions in the construction of anti-RPAS strategies.”

Discussion should be supplemented with the limitations of the study, as well as with some theoretical and practical (managerial) implications (who and for what can use the results of the study).

R.

The discussion section has been updated. (Lines 884 – 893)

“We should emphasize that all preferences established in the study, are based on the perspectives and vision of officers that worked on the analysis of the case, presenting the vision of part of some military organizations of BN, and no exposure to the vision of all BN or other Brazilian military forces, defense departments or remote technologies companies. Align with some limitations of the study, we emphasize that the study brings the perception of decision-making restricted for constructions and evaluation of anti-RPAS strategies in the sensing environment for defense, were the change of decision-makers or political environment can provide the establishment of different variables, objectives, and alternatives, exposing a new hierarchical structure with a variety of new actions as favorable solutions.”

Again, from the future research directions it is not clear what was the main focus of the paper. The authors should provide more general future research directions, not only about the methodology but also about the problem. Future research directions should be interesting to most of the Journal readership.

R.

The conclusion section has been updated. (Lines 931 – 940)

“In this context, we concluded that a given model has the potential to be applied in several areas of study through its flexibility and adaptability of the variables in a high-level decision-making context. For future studies, we established the evolution of this study concerning the construction of other objectives and alternatives aligned with the sensing environment for anti-RPAS strategies in defense, on the other hand, we search for the application of the proposed model in other case studies, looking to clarify points of improvement in the model and increase its robustness. Additionally, developing a computational platform for online access will be worked on, integrating several decision-makers in a dynamic evaluation environment, and operating in real-time, exploring numerical and graphical resources, thus supporting better decision-making.”

There are certain technical issues.

R.

All technical issues has been corrected on the new template.

Other improvements have been made as suggested by the other reviewers.

Reviewer 2 Report

1- The article is very long and has many redundant and repetitious sections. Shorten the text by removing repetitive topics and removing additional explanations; For example, the reference to the complexity or uncertainty of the problem is repeated in the first and second sections, and there are additional explanations regarding multi-Attribute decision-making approaches (such as compensatory and non-compensatory models), the group decision-making methods section has additional explanations, elements analysis You mentioned at least twice (before and after Figure 3), equations 1 and 2 are the same and...

2- In the introduction section (and a few other places in the text), you mentioned the issue of uncertainty in choosing a strategy; However, you have not provided any solution in this regard. It is better to mention this point in the discussion section or the result section (at least for future works).

3- Modify Figure 3. Some parts cannot be read.

4- In the ordinal evaluation with multiple decision-makers section, you mentioned Table 1, which is irrelevant.

5- In equation 3, does M represent the number of decision-makers? what is e?

6- Figure 5 should be adjusted.

7- You must provide valid references for all applied equations.

8- Add figures 7 and 8 (which are the most important figures in the article) to the text.

9- In the case study section, the last paragraph, I don't understand what you mean from the end of the paragraph!

"Considering that the analysis of criteria and alternatives is restricted to an operational evaluation, this study will restrict the evaluation to sensing technologies".

Please explain.

10- In the last paragraph of the case study section, what is the difference between levels 2 and 3?

11- The first paragraph of the section "Evaluation on Strategical and Tactical levels" does not match the last paragraph of the case study "In this context, the evaluation will consist of nine decision-makers, one decision-maker analyzing the first level of the hierarchy concerning the strategic objective, three decision-makers performing the analysis of the elements of the second level, two decision-makers evaluating the third level, and finally, three decision-makers evaluating the operational area of the process, consisting of criteria and alternatives so that each decision-maker will assess the operational part of their area of expertise".

Please explain.

12- For me, the most interesting part of your work is the combination of hard and soft OR approaches. However, neither in the methodology section nor in the case study section, you have made no clear explanations on how to use the soft approach. This is my most serious criticism of you. I suggest that by summarizing the text, dedicate a part of it to the soft approach.

13- Finally, you have mentioned uncertainty many times in the text. However, you have no suggestion to deal with it. I suggest that you pay more attention to this topic by reviewing the following works:

Sorourkhah, A., Babaie-Kafaki, S., Azar, A., & Shafiei Nikabadi, M. (2019). A Fuzzy-Weighted Approach to the Problem of Selecting the Right Strategy Using the Robustness Analysis (Case Study: Iran Automotive Industry). Fuzzy Information and Engineering, 11(1), 39–53. https://doi.org/10.1080/16168658.2021.1886811

Sorourkhah, A., & Edalatpanah, S. A. (2022). Using a Combination of Matrix Approach to Robustness Analysis (MARA) and Fuzzy DEMATEL-Based ANP (FDANP) to Choose the Best Decision. International Journal of Mathematical, Engineering and Management Sciences, 7(1), 68–80. https://doi.org/https://doi.org/10.33889/IJMEMS.2022.7.1.005.

14- Correct the highlighted parts in the text.

Author Response

PUNCTUAL RESPONSE TO THE REVIEWER 2

Rio de Janeiro, January 12, 2023.

Dear Reviewer of the Processes Journal,

First, we would like to thank you for all attention and contribution for the study. In this way, we describe below the treatment of the main changes proposed by the reviewers for the manuscript "SAPEVO-H² a Multi-Criteria Systematic Based on Hierarchical Structure: Decision-Making Analysis for Assessing anti-RPAS Strategies in Sensing Environment".

Reviewer#2

The article is very long and has many redundant and repetitious sections. Shorten the text by removing repetitive topics and removing additional explanations.

R.

The introduction section has been updated, and equation (1) removed. (Lines 42 – 63)

“As discussed by Shortland, Alison, and Barrett-Pink (2018), decision-making in political and military environments involves different levels and areas, interconnecting strategic, tactical, and operational analysis in favor of a direction aligned with the objectives in a given problematic situation [7]. In addition, it is necessary to consider that decision-making in the political and military sphere is complex, where the given form of solution can generate influences not only in the military sphere but also impacts other areas of society [8].

In the scenario of high-level decision analysis and impact in complex environments, integrating multiple stakeholders to determine and analyze aspects relevant to the problem is common, enabling, from multiple perspectives, to allow a consensus in decision-making [9]. According to [10] with the involvement of multiple scenarios and circumstances, the increase in complexity in a given analysis becomes noticeable, with different points of view as to the importance or influence of a decision variable [11], although, it is necessary to consider it in favor of a substantial evaluation and greater assertiveness in the final decision.

Considering the scenario of technological advancement that has taken place in recent decades, the improvement of military technologies has provided the use of revolutionary capabilities by military forces [12]. In this context, the Remotely Piloted Aircraft System (RPAS), mainly due to its versatility, has been considered a promising and desirable alternative to traditional flights  [13]. The capabilities provided by RPAS range from deploying weapons in distant wars to tracking/monitoring surveillance missions, among many others [14].”

In the introduction section (and a few other places in the text), you mentioned the issue of uncertainty in choosing a strategy; However, you have not provided any solution in this regard. It is better to mention this point in the discussion section or the result section (at least for future works).

R.

The limitation has been increased in the discussion section. (Lines 853 – 863)

“In decision-making analysis, the set of utilities obtained in the variables of the decision context has the main influence on the suggestion of the final decision. In this context of the proposed model, the peer-to-peer evaluation is efficient, considering that it is not always simple to perform a global preference assignment of a variable as a whole [121], thus, the clarification of global preferences is provided through a sequence of preference assignments between only two variables. Additionally, in the given model that was still proposed in its axiomatic structure, the feasibility of testing the consistencies of the attributions performed in the evaluations clarified the consistency relationships in the sets of preferences designated in the assessments. As a limitation of the model, we set that the model still does not present a treatment for uncertainty, in this way, we search to work in this gap in future studies, for more robustness of the model.”

Modify Figure 3. Some parts cannot be read.

R.

The figure 3 has been updated.

In the ordinal evaluation with multiple decision-makers section, you mentioned Table 1, which is irrelevant.

R.

The text has been updated, the correct was table 2.

In equation 3, does M represent the number of decision-makers? what is e?

R.

The e is the degree of importance for each element, and m the decision-maker. In this way, the text has been updated. (Lines 364 – 369)

“With the degrees obtained for each element , is performed the sum (2) for each decision-maker . Following the procedure, is obtained the normalization of the sums (3), indicating the respective importance of the elements for the attribute of the higher level. In this model, keeping the technique indicated in [25], if any criterion or attribute presents zero importance, is assigned for this element, 1% of the smallest value greater than zero.”

Figure 5 should be adjusted.

R.

The figure 5 has been updated.

Add figures 7 and 8 (which are the most important figures in the article) to the text.

R.

The text has been updated. (Lines 535 – 541 and 563 – 573)

“In a general context, through the interactions, a set of objectives were defined, which can be divided into strategic objectives, means, and ends for a given problematic situation [117]. For reasons of studies, this article will be restricted to the partial analysis of these established objectives, focusing only on objectives directly linked to the scenario of anti-SARP strategies. The causal map shown in Figure 7 presents a succinct demonstration of the constructed objectives network based on the techniques of the causal maps approach so that a given network will be later ranked based on the prioritizations of these objectives.”

“With a given analysis, one seeks not only to clarify the most favorable alternatives and prioritize the objectives on a global level but also to expose their respective performances at a local level, in line with the element of the higher level. In this context, the evaluation will consist of nine decision-makers, one decision-maker analyzing the first level of the hierarchy concerning the strategic objective, three decision-makers performing the analysis of the elements of the second level, two decision-makers evaluating the third level, and finally, three decision-makers evaluating the operational area of the process, consisting of criteria and alternatives, so that each decision-maker will assess the operational part of their area of expertise. Considering that the analysis of criteria and alternatives is restricted to an operational evaluation, this study will restrict the evaluation to sensing technologies. Figure 8 presents all details of hierarchical structure.”

In the case study section, the last paragraph, I don't understand what you mean from the end of the paragraph.

"Considering that the analysis of criteria and alternatives is restricted to an operational evaluation, this study will restrict the evaluation to sensing technologies".

R.

In this paragraph we try to present a delimitation of this study, evaluating just sensing technologies, in the last level of the hierarchical model, not considering combat Technologies as an example.

In the last paragraph of the case study section, what is the difference between levels 2 and 3?

R.

No, the analysis is the outranking conclusion concerning the elements of the first level. For a better understanding of the consideration, the text has been updated.

The first paragraph of the section "Evaluation on Strategical and Tactical levels" does not match the last paragraph of the case study "In this context, the evaluation will consist of nine decision-makers, one decision-maker analyzing the first level of the hierarchy …".

R.

The study is composed by nine decision-makers, six analyzing the first three levels, with objectives, then the other three decision-makers, analyze the other levels, concerning criteria and alternatives (technologies).

For me, the most interesting part of your work is the combination of hard and soft OR approaches. However, neither in the methodology section nor in the case study section, you have made no clear explanations on how to use the soft approach. This is my most serious criticism of you. I suggest that by summarizing the text, dedicate a part of it to the soft approach.

R.

The case study section has been updated. (Lines 498 – 512)

“Regarding the development of anti-SARP strategies, first, it is necessary to obtain a better understanding of the scenario of technologies currently present in Brazil, specifically, in the Brazilian Navy. As support in the understanding of this problematic situation, is used the construction of a causal map, which has as the basis of the methodological concepts of the SODA, from Soft OR, thus bringing a structuring of the scenario under analysis and construction of the objectives to be achieved with a given decision making. As presented by Abuabara and Paucar-Caceres [116], there are many studies related to the applications of SODA methodology in case studies based on scenarios of strategic analysis.

The SODA methodology was proposed by [26] and is based on the study of the situation in the form of a cognitive map, seeking to reflect the points of view of each member regarding the resolution of the problem situation, favoring the interaction between those involved in the process of decision analysis functioning as a facilitating device for obtaining consensus among the team's actors and commitment regarding the measures that should be taken.”

Finally, you have mentioned uncertainty many times in the text. However, you have no suggestion to deal with it. I suggest that you pay more attention to this topic by reviewing the following works.

R.

  1. Thank you for the suggested studies, both providing a robustness for the reference set.

Other improvements have been made as suggested by the other reviewers.

Reviewer 3 Report

Please read the attachment. Thank you. 

Author Response

PUNCTUAL RESPONSE TO THE REVIEWER 3

Rio de Janeiro, January 12, 2023.

Dear Reviewer of the Processes Journal,

First, we would like to thank you for all attention and contribution for the study. In this way, we describe below the treatment of the main changes proposed by the reviewers for the manuscript "SAPEVO-H² a Multi-Criteria Systematic Based on Hierarchical Structure: Decision-Making Analysis for Assessing anti-RPAS Strategies in Sensing Environment".

Reviewer#3

A well-structured scientific article should communicate research results and provide in-depth data analysis. The Reviewer suggests the authors have an outline for this paper. Please revise the structure of your paper correctly.

R.

All the template has been updated.

Figure 8 and the annotation for figure 8 Should be on 1 page if possible. The authors should resize or rearrange Figure 8 for better reading.

R.

The figure 8 has been updated.

Figures’ names: The figure titles should be put below the figures.

R.

The structures of titles have been updated.

A literature review for Multi-Criteria Selection Model and security with wireless communications needs to be added. The following work could match those gaps: Wireless Communications for Data Security: Efficiency Assessment of Cybersecurity Industry—A Promising Application for UAVs; A Two-Stage Multi-Criteria Supplier Selection Model for Sustainable Automotive Supply Chain under Uncertainty.

R.

Thank you for the suggested studies, both providing a robustness for the reference set.

Citations in the text and references should be followed the journal template. This research needs more references. The Reviewer suggests that you should search the Journal of Processes or other Journals for more references that could be used to enrich your literature review.

R.

The citations have been rearranged.

How did the authors evaluate the validity of their results?

R.

Considering that it is a multi-criteria modeling, the study in question was not based on an optimality process, but rather clarifying the most favorable objectives, actions and alternatives in the decision-making scenario, which is structured in a hierarchy, thus integrating multiples, explaining the results in a transparent way regarding the perception of the decision makers, individually and aggregated.

What are the main limitations of this approach.

R.

The main limitations have been updated in the discussion section (Lines 853 – 863 and 884 – 893)

“In decision-making analysis, the set of utilities obtained in the variables of the decision context has the main influence on the suggestion of the final decision. In this context of the proposed model, the peer-to-peer evaluation is efficient, considering that it is not always simple to perform a global preference assignment of a variable as a whole [121], thus, the clarification of global preferences is provided through a sequence of preference assignments between only two variables. Additionally, in the given model that was still proposed in its axiomatic structure, the feasibility of testing the consistencies of the attributions performed in the evaluations clarified the consistency relationships in the sets of preferences designated in the assessments. As a limitation of the model, we set that the model still does not present a treatment for uncertainty, in this way, we search to work in this gap in future studies, for more robustness of the model.”

“We should emphasize that all preferences established in the study, are based on the perspectives and vision of officers that worked on the analysis of the case, presenting the vision of part of some military organizations of BN, and no exposure to the vision of all BN or other Brazilian military forces, defense departments or remote technologies companies. Align with some limitations of the study, we emphasize that the study brings the perception of decision-making restricted for constructions and evaluation of anti-RPAS strategies in the sensing environment for defense, were the change of decision-makers or political environment can provide the establishment of different variables, objectives, and alternatives, exposing a new hierarchical structure with a variety of new actions as favorable solutions.”

Other improvements have been made as suggested by the other reviewers.

Round 2

Reviewer 1 Report

The authors have invested substantial effort to address all issues from the previous review round, thus significantly improving the quality of their paper. I suggest the acceptance of the paper in its present form.

Author Response

Rio de Janeiro, January 17, 2023.

Dear Reviewer of the Processes Journal,

First, we would like to thank you for all your attention and contribution to the study. All the improvements of the manuscript "SAPEVO-H² a Multi-Criteria Systematic Based on Hierarchical Structure: Decision-Making Analysis for Assessing anti-RPAS Strategies in Sensing Environment", were made possible by your direction in the review.

Best Regards.

Miguel Lellis

Reviewer 2 Report

I would like to thank the authors for their attention in revising the article. In addition to addressing my concerns well, they have edited the document with great care. Consequently, I consider the paper to be ready for publication and acceptance.

Author Response

(The authors gave the same response as above.)

Reviewer 3 Report

Please read the attachment. Thank you.

Author Response

PUNCTUAL RESPONSE TO THE REVIEWER 3

Rio de Janeiro, January 17, 2023.

Dear Reviewer of the Processes Journal,

First, we would like to thank you for all attention and contribution for the study. In this way, we describe below the treatment of the main changes proposed by the reviewers for the manuscript "SAPEVO-H² a Multi-Criteria Systematic Based on Hierarchical Structure: Decision-Making Analysis for Assessing anti-RPAS Strategies in Sensing Environment".

Reviewer#3 – 2º Round

The authors tried to answer my question. However, many contents still need to be added to the manuscript. Add what the authors present in the point-to-point response to reviewers to the author's manuscripts. Please specify where the authors have updated the information.

R.

Considering all changes provided in the first round of review, attached follows a pdf highlighting all changes.

Please add the research gaps. Why are there so many new and popular MCDM methods that the authors do not use, using this method? Decision-making is only a trade-off. Whether the academic significance of this paper is reliable for practical application is still a subjective opinion. Let's argue further to convince the reviewer that these contributions are necessary for humanity.

R.

Regarding the gaps of study, the introduction section has been updated, exposing now the gaps and motivation for study. (Lines 81 – 91)

“As presented in [23], in real problematic situations of decision analysis, uncertainty variables are intrinsic to the assessment [24], especially in environments involving multiple decision-makers belonging to different levels of strategic, where members generally disagree about the parameter values, and preference assignments [25]. Nevertheless, even though group decisions are expected in analyzing real situations, few models consider the formation of subgroups to deal with problems at strategic, tactical, and operational levels in particular [26], in other words, most of the models do not consider hierarchical analysis, providing the integration of different levels of expertise in the decision-making process, being this problematic situation a motivation of the methodology development, trying to fill this gap regarding group decision-making analysis in complex scenarios.”

Please re-edit the manuscript appropriately in English style.

R.

All text grammar has been revised.

Please make a table or chart to compare your proposed results with others.

R.

Many thanks for this contribution in specific, a descriptive and comparative table regarding classic models has been included in the discussion section, along with the addition of a new paragraph. (Lines 861 – 870)

“In a complementary way, based partially on the approach presented in [127], some limitations and comparisons are necessary, thus exposing some general characteristics of viability of the proposed model compared to the traditional models already existing in the literature. One of the major limitations identified as the motivation for the developed method concern the consideration of only two levels of evaluation, established as criteria and alternatives, although, as already demonstrated in this study, in many real cases, it is necessary to include other levels , thus transcribing the necessaries objectives to reach the final decision-making. In this way, table 29 exposes the main characteristics of the proposed model, regarding the main models present in the literature, listing limitations, feasibility and respective particularities of all models discussed.”

Round 3

Reviewer 3 Report

Dear Editor and Authors: 

Thank you for providing the point-to-point response. 

The authors have carefully and patiently corrected and answered all comments and questions. The manuscript has been improved and looks perfect now. The reviewer strongly suggests it be accepted for publication. 

(only one minimal concern: please format the last reference as the guidance of the journal template).

Thank you for reading.

Sincerely yours, The Reviewer.